# Sequence terminus dependent PCR for site-specific mutation and modification detection

Gaolian Xu[1,3], Hao Yang[1,3], Jiani Qiu[1,3], Julien Reboud [2], Linqing Zhen[1], Wei Ren[1], Hong Xu[1] ✉, Jonathan M. Cooper [2] ✉ & Hongchen Gu[1] ✉

The detection of changes in nucleic acid sequences at specific sites remains a critical challenge in epigenetics, diagnostics and therapeutics. To date, such assays often require extensive time, expertise and infrastructure for their implementation, limiting their application in clinical settings. Here we demonstrate a generalizable method, named Specific Terminal Mediated Polymerase Chain Reaction (STEM-PCR) for the detection of DNA modifications at specific sites, in a similar way as DNA sequencing techniques, but using simple and widely accessible PCR-based workflows. We apply the technique to both for site-specific methylation and co-methylation analysis, importantly using a bisulfite-free process - so providing an ease of sample processing coupled with a sensitivity 20-fold better than current gold-standard techniques. To demonstrate the clinical applicability through the detection of single base mutations with high sensitivity and no-cross reaction with the wild-type background, we show the bisulfite-free detection of *SEPTIN9* and *SFRP2* gene methylation in patients (as key biomarkers in the prognosis and diagnosis of tumours).

The recognition of site-specific mutations, deletions, and modifications of nucleic acids is increasingly important as these sequences become validated as clinical biomarkers for a wide range of diseases, including cancer[1,2] and neurological disorders[3]. The recent acknowledgement that such site-specific changes carry significant genomic and epigenetic information, has helped advance the development of new methods for the analysis of nucleic acids with a single base resolution, using nanopore[4] and single molecular real-time[5,6] sequencing. These techniques have been adapted to enable the detection of site-specific changes, including both single base mutation and 5-mC methylation[4–6]. However, the protocols remain complex to implement, limiting their applications in the clinic. For example, in order to avoid bias during the creation of the libraries, there is the need for extensive bioinformatics analysis, including base calling, sequence alignment, and statistical analysis[7].

In contrast, the amplification of DNA sequences using polymerase chain reaction (PCR) is now integrated into many routine clinical analysis workflows, for example, for infectious disease diagnostics. Currently, however, these methods are not able to identify site-specific mutations at single-base level, nor to detect epigenetic base modifications, leading to a number of enhanced or modified PCR protocols, including allele-specific (AS) quantitative (q)PCR (see Supplementary Note 1 for Glossary), which uses blocking oligonucleotides to enhance the performance of site-specific mutation analysis[8]. In such cases, the mutated target sequence is amplified selectively using a specific AS primer, in which the nucleotide substitution site is located at the 3′ end, requiring detailed knowledge about the location a priori. The complexity of the protocol limits both the number of mutations that can be analysed, and the sensitivity of the analysis[9].

AS-qPCR can also be used to identify allele-specific methylation using bisulfite pretreatment[10], but again, the accuracy of the detection has been shown to be low for different modification states[8]. Thus, in order to mitigate the limitations of AS-qPCR for single base mutation analysis, linked to the need for specific AS primers, restriction enzyme

[1]School of Biomedical Engineering/Med-X Research Institute, Shanghai Jiao Tong University, Shanghai 200030, China. [2]Division of Biomedical Engineering, University of Glasgow, G12 8LT Glasgow, United Kingdom. [3]These authors contributed equally: Gaolian Xu, Hao Yang, Jiani Qiu. ✉ e-mail: xuhong@sjtu.edu.cn; Jon.Cooper@glasgow.ac.uk; hcgu@sjtu.edu.cn

(RE) digestion has been integrated into PCR workflows to identify site-specific changes (including, for example, end-point PCR-restriction fragment length polymorphism (PCR-RFLP)[11] for single base mutation, and methylation-sensitive restriction enzyme (MSRE)-treated PCR[12] for DNA methylation analysis). Whilst RFLP is limited to low sensitivity and requires labor-intensive processing, MSRE-treated PCR is able to provide information on the region of the modifications, by comparing the amplification efficiency between RE-treated and untreated samples. However, the technique requires both MSREs and isoschizomers simultaneously, which not only limits the range of sequences that the method can be used for[12] but also requires extensive process optimization to ensure that the digestion is complete, as even traces of undigested DNA can lead to false-positive results[13]. The amplification of only the digested product can overcome these limitations (as performed in a helper-dependent chain reaction (HDCR)[14] and ligation-mediated PCR[15]); however, the efficiency is low, leading to low sensitivity and limited applications.

In order to overcome such complexities and analytical shortcomings, here we demonstrate a simple and generic PCR-based strategy to identify nucleic acid modifications and single base mutation, with high specificity, enabling the site-specific location of any sequence variation to be identified with single base resolution. The method, named Specific Terminal Mediated PCR (STEM-PCR), has the potential to provide similar levels of information as sequencing techniques, but with a step change in simplicity, utilizing routine PCR systems, to unlock the wider clinical value of genomic and epigenetic data.

STEM-PCR relies on a simple treatment of the target sequences, which generates a molecular construct that carries specific terminal parts (Fig. 1a) and the ability to self-fold and initiate PCR primer binding for amplification. The technique thus only requires a simple toolbox of DNA sequences and a specific design mechanism for the formation of the complex (determined a priori, depending on the modification that is being detected). The only treatment methods required are those to generate a specific terminus, such that they can be chosen from a wide range of options, including gene editing tools, primer hybridization, or template strand with specific ends.

To demonstrate the potential impact of the STEM-PCR strategy, we focus on DNA methylation and mutation as case studies with important practical clinical implications. Since REs-based digestion can be performed using mild conditions, whilst achieving high specificity, and is compatible with the subsequent qPCR process, we design three different RE-based assays with high clinical relevance, to detect DNA methylation, including those for multiple sites simultaneously at a time. We illustrate this capability through the design of a methylation-dependent restriction endonuclease (MDRE)-based single site bisulfite-free assay (greatly simplifying sample preparation protocols). We also demonstrate the detection of single base mutation/deletion with peptide nucleic acid (PNA) extension blockers.

These examples demonstrate the ability of STEM-PCR to be adapted to detect different modifications with a single, simple strategy, in a generalizable approach, where existing techniques require completely different mechanisms, and their subsequent optimizations. In order to show the breadth of clinical applicability, we present results from patient samples enabling the identification of single base mutations with high sensitivity and no-cross reaction with the wild-type background for the bisulfite-free detection of *SEPTIN9* and *SFRP2* gene methylation (both as key biomarkers in the understanding of prognosis and diagnosis of a range tumors, including colorectal, breast, and prostate cancers).

## Results

### Concept mechanism of STEM-PCR

We demonstrate that the specificity of STEM-PCR is based on the generation of a self-folding, self-priming, hairpin-based template nucleic acid sequence with site-specific ends, which can conveniently be achieved with a range of biological or chemical treatments (Fig. 1a). The template subsequently initiates a PCR amplification, mediated through a primer group (Fig. 1b) that targets the specific end modification.

We first demonstrated the underlying concept for site-specific detection with the STEM-PCR strategy using RE digestion as a proof of principle (Fig. 1c). When the sequence recognized by the enzymes is present, the digestion leads to the generation of the sequence P1 with a specific and defined 5′ end, while the undigested target remains intact (Fig. 1c). In the amplification step, P1 acts as a template and initiates a linear strand synthesis, via a tailored-designed foldable primer (TFP), which hybridizes with both the digested and undigested templates. For the digested molecule (P1), this stops at the 5′ end, generating a construct that can self-fold (P2) and self-prime into a complete hairpin structure (P3), without 3′ end overhang, whilst for the undigested molecule, the synthesis continues, preventing self-priming (P4).

The TFP requires at least four distinct components: a capture region (CR) that is complementary to the target sequence at the 3′ end; an artificial primer (APs) sequence at the 5′ end, which is independent of the template sequence and can thus be designed to maximize binding and thermodynamic efficiency during the followed amplification process; a folding region (FR), which is the same as the 5′ end of P1; and an extension blocker between CR and FR (Fig. 1b). The complete hairpin structure (P3) initiates the exponential amplification process using the APs.

### Optimization of STEM-PCR

In contrast to conventional PCR amplification, the innovation underpinning STEM-PCR's ability to identify sequence-specific information is based upon a mechanism involving the generation of an intermediate structure designed through the different sequence components of the treated template and the TFP. In particular, the FR sequence is important for the stability of the structure of P3, and we show that as the FR length was reduced from 20 to 8 bases, the stability of the hairpin decreased (as measured by the melting temperature Tm in Fig. 1d and thermodynamically with ΔG in Supplementary Fig. 1). The impact of the length of FR on stability translates into higher amplification efficiency, as measured by $Ct$ values (Fig. 1e). The lowest template concentration leading to amplification shifted from $10^5$ copies/reaction when FR was 6/9 nt long, to $10^4$ copies/reaction at 12 nt. When considering the evolution of $Ct$ for a single concentration of $10^5$ copies/reaction, Fig. 1e shows that increasing FR length yielded faster amplification (lower $Ct$), until 18 nt, after which the number of P3 molecules formed during P2 self-folding and self-priming did not increase further.

The efficiency of the amplification of STEM-PCR is linked to a competition between P3 self-folding and hetero-hybridization with AP (Supplementary Fig. 2). Formally, within the kinetic framework of the reaction as a two-state approach (Equation 9 in Supplementary Note 2), this is illustrated by the fact that, as the number of AP molecules increases, the number of self-folded P3 and single strand P3 decreases leading to more hetero-hybridized complexes. This structure can also be controlled by the stem/loop ratio in P3, which can increase stability and, consequently, amplification efficiency, yielding lower $Ct$ values for longer structures (Supplementary Fig. 3).

### STEM-PCR detects single-site methylation

STEM-PCR is able to use different REs flexibly to generate the specific intermediary constructs depending on the modifications studied (Fig. 1a), including, for example, GlaI ($R^mCGY$), FspEI ($C^mC(N)_{12}$), and LpnPI ($C^mCDG(N)_{10}$). We used MspJI ($^mCNNR(N)_9$) as an example[16] with four different artificial templates (T1-T4 in Supplementary Table 1) with the same sequence but different CpG modifications (Fig. 2a, primers in Supplementary Table 1). The results from gel electrophoresis analysis

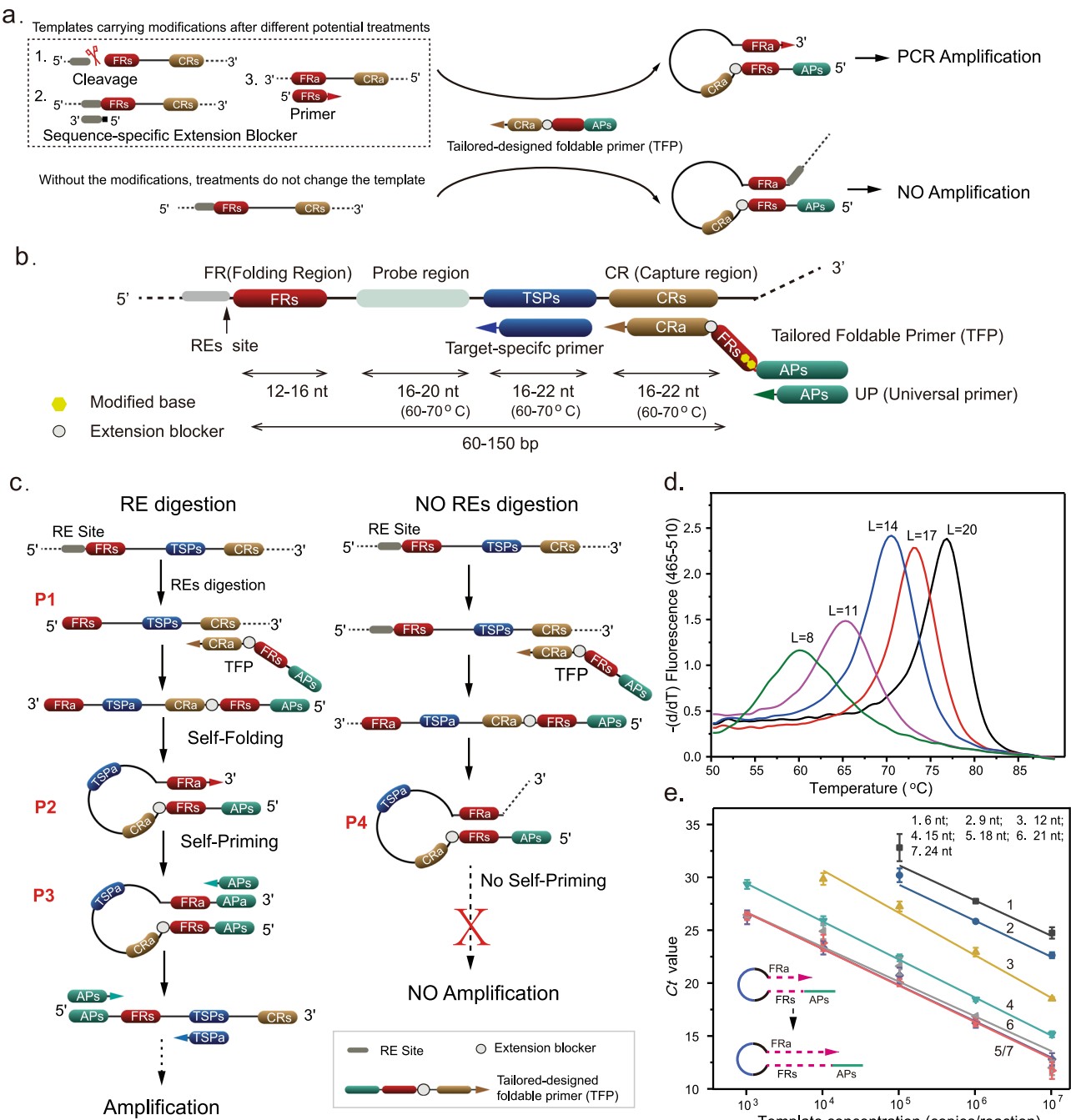

**Fig. 1 | STEM-PCR for location-specific methylation detection. a** The mechanism of STEM-PCR with different template strategies. The extension blocker is PEG18, equivalent to 6 nt in length[31], noted as iSp18 in Supplementary Tables 1–7. **b** STEM-PCR amplification primer design showing the recommended distance between primer regions and a melting temperature (Tm) range for each primer. Primers are indicated by solid arrows and given the corresponding capital letter designation. **c** The mechanism of STEM-PCR for RE digestion mediated location-specific detection. The lower-case letter s/a indicates the sense and antisense of the sequence; **d** DNA melting for different sequence lengths, L, of the stem from 8

(green), 11 (purple), 14 (blue), 17 (red), to 20 nt (black), with the same loop size (40 nt); **e** The Ct values as a function of 10x serial diluted hairpin structures, from $10^7$ to $10^3$ copies/reaction, for different FR lengths of P1 from 6 to 24 nt with the same primer set and loop size (40 nt). Data were the average of three replicates and error bars represent the standard deviation. All linear regressions have $R^2 > 0.95$. (6 nt – 0.95, black; 9 nt – 0.98, blue; 12 nt – 0.99, yellow; 15 nt – 0.99, green; 18 nt – 0.99, dark blue; 21 nt – 0.99, gray; 24 nt – 0.99, red). Source data are provided as a Source Data file.

after amplification (Fig. 2b) demonstrate that STEM-PCR is specific to a chosen sequence (identified by the helper DNA fragments required for the enzyme's functionality), even when the neighboring CpG modification is inserted within one base (Supplementary Fig. 4 and Supplementary Table 2 for sequences).

To demonstrate the versatility of the STEM-PCR approach, we used a second enzyme, GlaI, to detect a methylated site within the

human *SEPTIN9* gene (chr17: 7737351824), a promising biomarker for colorectal cancer (CRC)[17]. The methylation was confirmed by pyrosequencing using a standard bisulfite pretreatment procedure (Supplementary Fig. 5), where the methylated target was serially diluted amongst unmethylated background (10, 1, and 0.1% in 10,000 copies of unmethylated DNA). Results analysed by gel electrophoresis (Fig. 2c) demonstrate the sensitivity of STEM-PCR and show that the

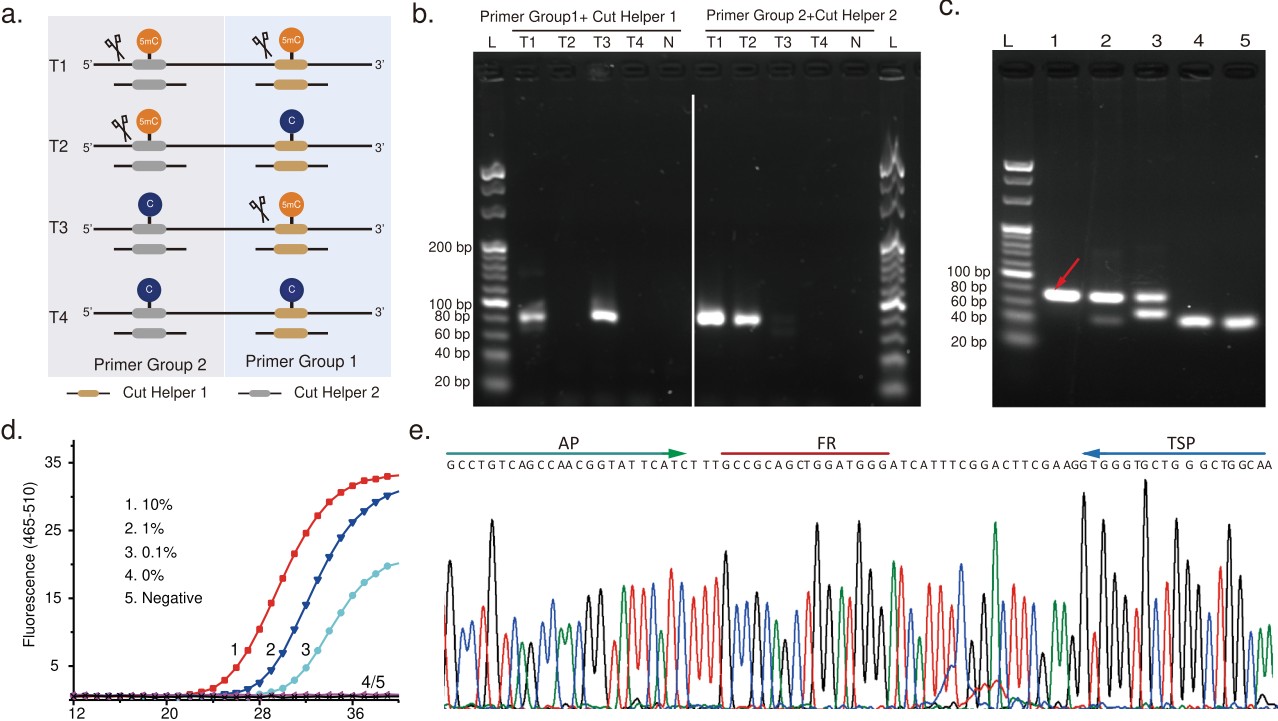

**Fig. 2 | Location-specific detection of methylation.** Location-specific detection of methylation (**a**) scheme describing the mechanism used for location-specific detection of 5-mC; **b** An agarose gel showing the amplification products generated with different cut helper sequences (required for the functionality of the enzymes) and location-specific primer sets. Four artificial templates were designed with the same sequence but contained two methylated CpG sites (distance = 20 bp), which can be digested using MspJI (mCNNR). L, 20 bp ladder; N, no template, as a negative control. Cut helper 1 and primer sets 1 were designed for the detection of the right target CpG site; cut helper 2 and primer group 2 were designed for the left one shown in (**a**); each experiment was repeated independently at least three times with similar results (**c**) STEM-PCR products of ten times series diluted GlaI-cut methylated *SEPTIN9* with 10,000 copy unmethylated background separated by agarose gel electrophoresis (1). 10% (2).1% (3). 0.1 (4). 0% (5). Negative control L, 20 bp ladder. The red arrow indicates the expected amplicon (79 bp), with the bottom band as a primer-dimer (40 bp). Each experiment was repeated independently at least three times with similar results; **d** Typical amplification curves of ten times series diluted GlaI-cut methylated *SEPTIN9* with 10,000 copy unmethylated background using STEM-PCR—only one sample is shown per condition for ease of viewing (red – 10%, blue – 1%, light blue – 0.1%, purple – 0%, black – negative); **e** The bands corresponding to STEM-PCR product (red arrow in **c**) were excised from the gel, ligated into a plasmid, and then sequenced, showing the structures described in Fig. 1a. Source data are provided as a Source Data file.

assay can detect the methylation biomarker signature as a signal of 0.1% of the unmethylated background.

The specificity was demonstrated by challenging the method with hypermethylated DNA (from 1000 to 0 copies) within 10,000 copies of unmethylated background (Fig. 2d). Results showed no cross-reactivity to the unmethylated sequence and no loss of sensitivity (as 0.1% methylated template could be detected). Sequencing of the band corresponding to STEM-PCR products again outlined the presence of AP and TSP at the ends (Fig. 2e), indicating the mechanism and reaction steps of STEM-PCR performed as expected.

Figure 3a demonstrates that STEM-PCR is not only highly specific, but also sensitive, achieving a limit of detection of the methylation in *SEPTIN9* down to 15 pg/reaction (ca. 5 copies/reaction), which is ~20-fold better than with a standard HeavyMethyl assay (Supplementary Fig. 6a). The *Ct* value shows a linear dependence ($R^2 > 0.99$) for concentrations over three orders of magnitude of dilution, indicating the quantitative potential of STEM-PCR. The limit of detection was confirmed statistically with 20 replicates, to be lower than 5 copies/reaction (Fig. 3b). An implementation in a digital PCR format yielded single copy sensitivity (Supplementary Fig. 6b) and confirmed the potential for quantification, which could be of interest in therapeutic monitoring[18].

Further, to test the feasibility of STEM-PCR with lower quality DNA (as might be expected with library or reference samples stored for long periods of time), we processed ten formalin-fixed paraffin-embedded (FFPE) samples, which were >10 years old. After extraction with a commercial kit (TAOGEN Inc., yielding concentrations 10.5–29.5 ng/µl; 260/280 ratio 1.94–2.12), results from STEM-PCR enabled differentiation between cancer types (Supplementary Fig. 7), illustrating the potential for the technique to be used in retrospective studies, and with a wide range of samples.

We further compared the performance of STEM-PCR to benchmark technique Bisulfite-PCR (BS-PCR) sequencing[19], using formalin-fixed paraffin-embedded tissue samples from 20 individuals undergoing treatment for cancer, to detect methylated sites in *SEPTIN9* and *SFRP2* genes (chr4:153788844)[20], with sequences listed in Supplementary Table 3. Figure 3e shows that STEM-PCR provided the same information as BS-PCR sequencing for 35% (7/20) and 80% (16/20) of samples for *SFRP2* and *SEPTIN9*, respectively. Discordant results were further investigated using methylation-specific PCR (MS-PCR) sequencing (Fig. 3c), which revealed the existence of a methylated cytosine undetected by BS-PCR sequencing, suggesting that the sensitivity of STEM-PCR is better than that of standard BS-PCR sequencing. To clarify the robustness of these results, we further increased the sensitivity of BS-PCR, by amplifying the samples with low levels of methylation[21] (after being diluted 10,000 X, Supplementary Fig. 8, primers listed in Supplementary Table 4). Typical results from clinical sample sequencing for *SEPTIN9* and *SFRP2* confirm the limitations in the performance of BS-PCR (Supplementary Fig. 8)[19].

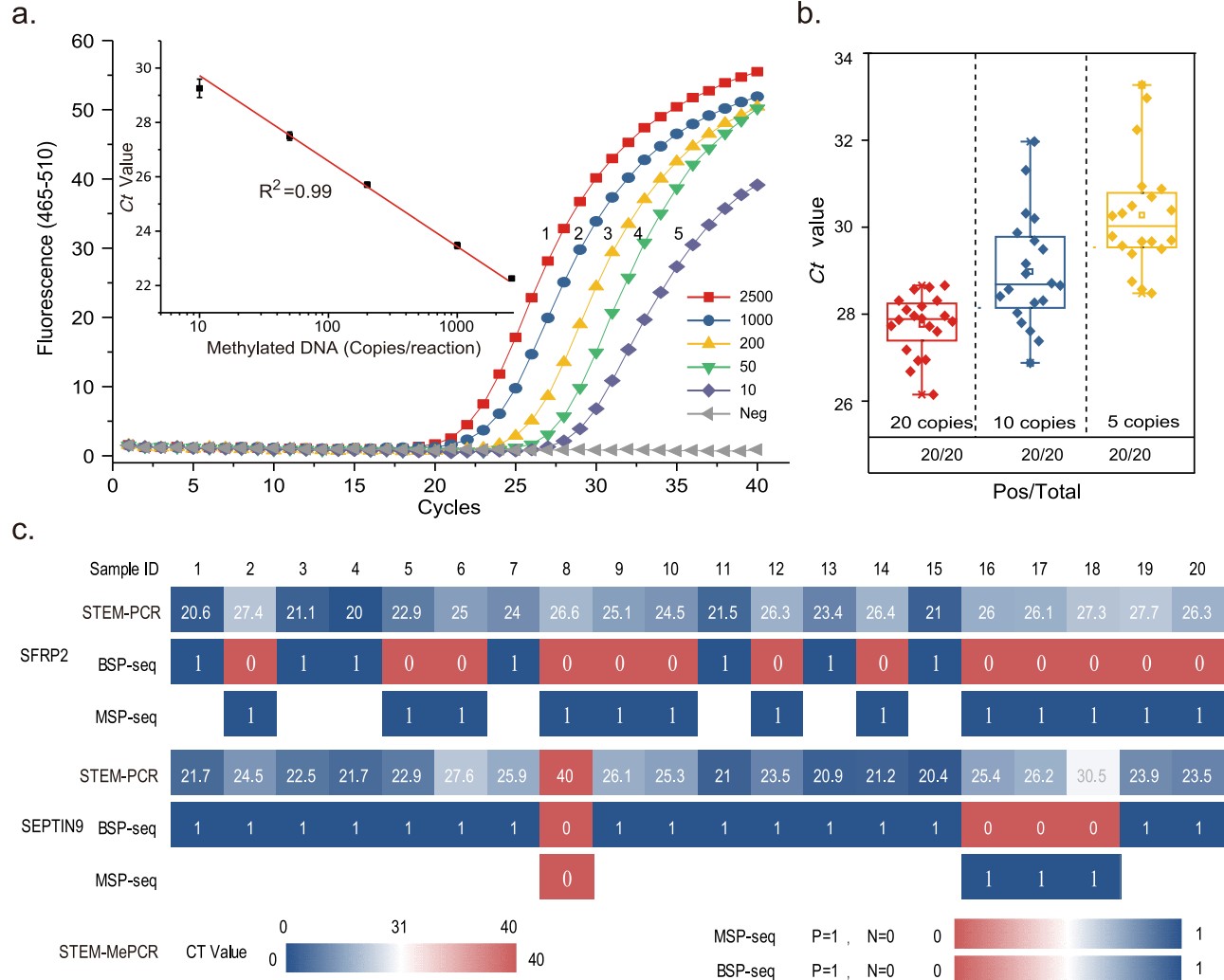

**Fig. 3 | Performance of STEM-PCR for methylation detection. a** Real-time amplification curve of STEM-PCR with serially diluted templates (copies/reaction). (1) 2500 (red squares); (2) 1000 (blue disks); (3) 200 (yellow triangles); (4) 50 (green inverted triangles); (5) 10 (turquoise lozenges); (6) ddH$_2$O as a negative control (gray triangles). $Ct$ value as a function of template concentration and fitted with linear regression ($R^2 = 0.99$) (inset). Data were the average of three replicates, and error bars represent the standard deviation; **b** The results of 20 independent experiments of STEM-PCR with 5 (yellow), 10 (blue), and 20 copies/reaction (red) input methylated DNA using GlaI on *SEPTIN9* genomic DNA−box plots display the average, boundaries are upper and lower quartiles respectively, whilst top and bottom bars are maximum and minimum (all points are shown); **c** Comparison of STEM-PCR and BSP-sequencing and MSP-sequencing[21]. Source data are provided as a Source Data file.

## STEM-PCR for single base mutation detection using PNA modifications

Although using RE digestion enabled us to compare STEM-PCR's performance to existing methods (which also rely on enzymes), it does require a specific sequence in the target to be recognized by the enzymes, in the same way as for other RE-based methods[14]. The underlying STEM-PCR mechanism can, however, enable a more generic approach, which we illustrate here with the use of PNA-modified constructs (Fig. 4a).

In this context, we used PNA modifications to enable us to discriminate single base mismatches as well as block polymerase progression[22], with the higher binding affinity of PNA oligonucleotides (as compared to their DNA analogs) biasing the competition between the STEM-PCR reaction components, stabilizing P1, whilst also inhibiting the extension process of TFP at the 5′ end of FRs to generate P2. As such, a single base mismatch prevents the hybridization of PNA oligonucleotide at the target sequence, which leads to the generation of P4 (with a long tail overhang after self-folding), and prevents amplification. The impact of the level of template fragmentation on

STEM-PCR was studied in Supplementary Fig. 9. When the truncated position site at the 5′ end of FRs was partly absent from the PNA blocker, this prevented the hybridization of PNA, which in turn led to the extension of TFP, generating a short tail overhang after self-folding, and preventing amplification (identical to that for untruncated wild-type sequences).

The STEM-PCR implementation with PNA showed similar high performance as its enzymatic counterpart, demonstrated with the detection of the L858R single base mutation in EGFR[23] (Fig. 4b), with a sensitivity of 30 copies/reaction and high specificity with no signal with the wild-type sequence, even at a concentration of 3000 copies/reaction, (primer information in Supplementary Table 5). The choice between the different implementations of STEM-PCR should also take into account the fact that REs digestion is more efficient[24] than the hybridization of the PNA blocker[25]. This effect is compounded by the fact that this lower efficiency is amplified at the stage of the synthesis of the full hairpin structure. Nevertheless, the specificity was as low as 1% mutated template detected in a background of 3000 copies of wild-type DNA (Supplementary Fig. 9b). Furthermore, since

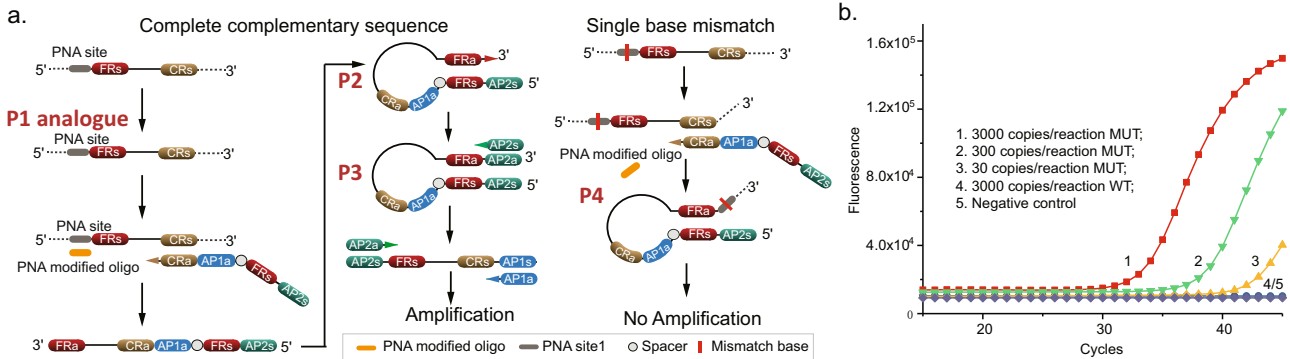

**Fig. 4 | STEM-PCR for site-specific detection with PNA-modified oligo. a** The mechanism of PNA-modified oligo-mediated STEM-PCR for single base mismatch detection; **b** The amplification results of L858R mutation using PNA-based STEM-PCR. (1) 3000 copies/reaction (red squares); (2) 300 copies/reaction (green down triangle); (3) 30 copies/reaction (yellow up triangle;) (4) 3000 copies/reaction of wild-type template (blue disks); (5) negative control (turquoise square). Source data are provided as a Source Data file.

the PNA blocker does not compete with TFP, STEM-PCR offers extensive design flexibility to different mutations. However, it should be noted that each mutation requires its own set of PNA blockers.

## STEM-PCR detects co-methylation

STEM-PCR can also be readily adapted to detect co-methylation, which has been shown to be important clinically[26], especially in circulating tumor (ct)DNA. The latter presents the added challenge of being highly fragmented and usually present in a high background of unmodified sequences[27]. The mechanism is scaled to multiple methylations, by simply providing additional target-independent primers sequences AP2a and AP3s to stabilize the reaction intermediate structures from the different sequences, as well as a bridge primer (BP), to link the modifications into the new TFP sequence (Fig. 5a). The 3' end of BP is blocked and complementary to the target sequence, with an artificial sequence (AP1a) as the 5' end sequence. When designing primers, the potential to form secondary structures should be avoided and coordinated melting temperature (Tm) values should be considered.

To demonstrate this, human genomic DNA was extracted from cultured cells and fragmented to a size of ca. 150 bp (Supplementary Fig. 10) as a model for methylated ctDNA. The samples were treated with MDREs simultaneously to generate the single strands with specific 5' and 3' ends (P1), while keeping the unmethylated DNA intact. The free 3' end of P1 hybridizes with the CRa sequence of BP, resulting in a linear extension introducing AP1s at the 3' end, a sequence that does not exist in the initial template. TFP binds to the linear extension product via AP1a, and extends to stop at its 5' end, thereby introducing FRa, which leads to the generation of P2. The formation of this intra-molecular construct is a critical step for STEM-PCR, since its folding and self-priming leads to the generation of P3 with a complete hairpin structure and no 3' overhang (Figs. 1a, 5a), which can then serve as a template for the exponential amplification (with AP3s and AP2s). For the unmethylated sequence, however, the absence of MDREs-mediated specific digestion prevented the BP-based linear extension due to the lack of a free 3' end. We note that different restriction enzymes can be used with different restriction functionalities and we provide four types of digestion scenarios using different type MDREs as examples (Supplementary Fig. 11).

We demonstrate the performance of STEM-PCR for co-methylation analysis using GlaI as the restriction enzyme on the *SEPTIN9* gene (chr17, 77373471-77373520), sequences listed in Supplementary Table 6. Figure 5b, c show the method's specificity, which was confirmed with sequencing of the reaction products (Fig. 5d). The method is robust to fragmentation (which can be random) except when the breaks occur at the restriction sites, as illustrated in Fig. 6a. When there is no break at the 3' end of P1, (scenario T1 and T2),

BP-mediated linear extension does not occur, although a single break at the 3' end (T3) starts linear extension, leading to a long 3' overhang after self-folding of P2, preventing the introduction of AP3a and consequently the amplification. Only the coexistence of specific 3' and 5' ends within a same strand (T4) leads to linear extension, the generation of P3 with a hairpin structure, and the final amplification. When using methylated *SEPTIN9* sequences, which contain both a LpnPI and GlaI site, only the sample treated with both MDREs yielded a specific amplicon band (Fig. 6b).

The impact of the level of fragmentation on co-methylation detection was also studied using different artificial templates. The results of different lengths of FRc (Supplementary Fig. 12a) indicated that the truncated position at the 5' end of FRs prevented STEM-PCR, even with only one single extra base. We hypothesize that this may have been due to the overhung bases not only preventing the extension of FRa after self-folding, but also leading to a mismatch of AP2s. When the truncated position sites were located within FRs regions, the amplification efficiency decreased with the decrease in the generation of FRa. For example, the efficiency was ca. 10 times lower when 2 and 3 bases were deleted from the standard FRs. A similar study was carried out to explore the effect of CRs (Supplementary Fig. 12b), showing again a decrease in efficiency as the length of CRs decreased. One base missed from the CRs leads to a five-fold reduced efficiency, whilst more bases missed prevented amplification completely.

We characterized the effect of the length of the template (between the two restriction sites) by creating different sequences P1 with different lengths, from 45–165 bp, but the same FRs and CRs sequences (primers listed in Table S7). The results shown in Fig. 6c indicate the *Ct* value increased with length, confirming a lower amplification efficiency with the increase of P1 length (Fig. 6c), but nevertheless, generating a detectable signal up to 165 bp. The sizes of the final amplicons were confirmed on agarose gel, (Fig. 6c-inset), which also shows that the amount of primer-dimer decreased with P1 length, due to the increased utilization of primers for more efficient reactions.

Figure 6d shows that the *Ct* value of the STEM-PCR reaction increased as the amount of serially-diluted hypermethylated fragmented DNA increased from 10 copies/reaction to 3000 copies/reaction, with GlaI digestion (Fig. 6d). The limit of detection for *SEPTIN9* ctDNA assay was estimated as ~10 copies/reaction. The methylation signal quantified by STEM-PCR was detected with a dilution as low as 0.5%, with no non-specific signal within a background of 10 ng fragmented unmethylated DNA (Fig. 6e), whilst the *Ct* value showed a linear relationship with the target amount. When the background was increased to 20 ng, only one out of 8 replicates generated a

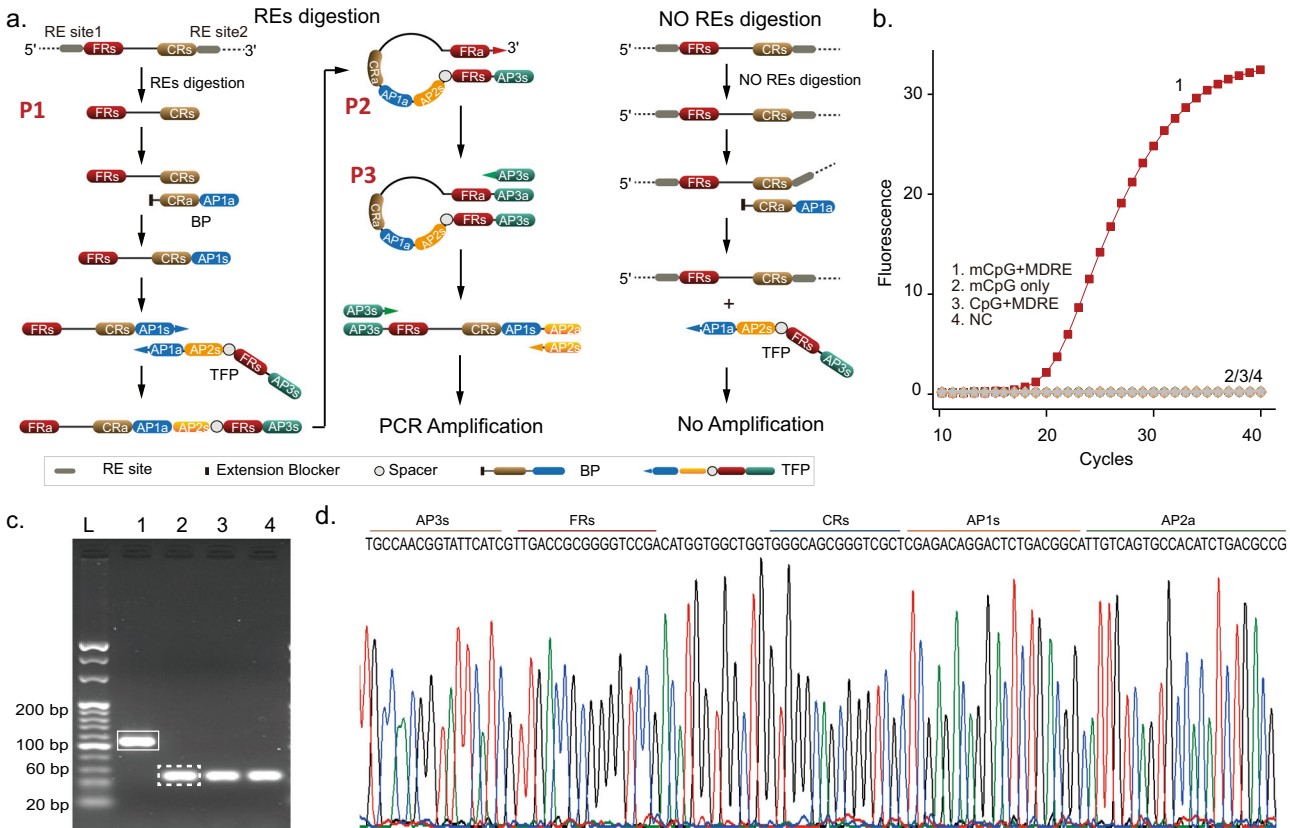

**Fig. 5 | STEM-PCR for co-methylation detection. a** Mechanism of STEM-PCR for cfDNA co-methylation detection; **b** Detection of GlaI-cut methylated *SEPTIN9*. (1) GlaI-cut methylated *SEPTIN9* - red; (2) Methylated *SEPTIN9* without GlaI – blue; (3) GlaI-cut unmethylated *SEPTIN9* - orange; (4) ddH₂O as a negative control - gray; **c** Agarose gel of GlaI-cut methylated *SEPTIN9* using STEM-PCR. L, 20 bp ladder. (1) GlaI-cut methylated *SEPTIN9*; (2) Methylated *SEPTIN9* without GlaI; (3) GlaI treated unmethylated *SEPTIN9*; (4) ddH₂O as a negative control. The band in the white box indicates the expected amplicon (109 bp). The band in a white dotted box indicates primer-dimer; each experiment was repeated independently at least three times with similar results. **d** Sequencing of STEM-PCR product. The band indicated by the white box in (**c**) was excised, cloned, and sequenced. Source data are provided as a Source Data file.

non-specific (false positive) result (Supplementary Fig. 13), indicating the specificity of STEM-PCR for highly fragmented ctDNA.

To evaluate the clinical utility of the mechanism of co-methylation detection described in Fig. 5, cfDNA extracted from 14 patients was tested using co-methylated STEM-PCR, and compared with gold-standard bisulfite sequencing, with the result shown in Supplementary Table 8. Two samples (12, 13) showed positive with STEM-PCR ($Ct$ value 27.86 and 25.25, equivalent to ca. 4 and 17 copies). The methylated results from bisulfite sequencing (Apogenomics Biotech CO Ltd, Shanghai, China) are also presented as methylation haplotype load[28], which was 1.89% (12) and 5.49% (13), whilst all other samples had measures below 0.5% (considered as negative, following the kit recommendations). STEM-PCR matched the performance of standard bisulfite sequencing, but with only 1 ng ctDNA and no bisulfite pre-treatment, confirming the potential of the technique to perform advantageously in applications where the source material is limited. However, a more extensive clinical evaluation will be required in the future, to establish the clinical limit of detection.

## Discussion

Genetic variants, in which sequences can be modified through deletion, mutations, or epigenetic changes at specific sites have been linked to a large number of diseases, including a number of cancers[29], yielding the development of methods and assays for diagnostic purposes (including, for example as ligation-PCR, PCR-RFLP, and HDCR). However, each approach has its own shortcomings, including its limited sensitivity, specificity, and/or the lengthy processing required. In contrast, our STEM-PCR strategy is based upon the generation of an

intermediate secondary nucleic acid structure, using a set of primer sequences that interact and hybridize specifically to a target sequence, generating constructs with defined end structures that can self-fold into hairpin structures and self-prime for PCR amplification.

In this work, we have demonstrated the potential of the approach, which does not use bisulfite preparation, with RE digestion, to detect single or multiple methylation sites, as well as PNA oligo-based hybridization for an enzyme-free assay. Furthermore, we note that our proposed principle is applicable to any strategy that generates the constructs with the required defined ends. The technique could readily be expanded to other approaches using gene editing tools, for example, CRISPR. We also use PCR as an amplification step, but we anticipate that other amplification reactions could be integrated into the same approach by modifying the constructs compositions. Potentially, in future, this could enable the technique to be implemented in completely enzyme-free processing, or using instrumentally simple, isothermal amplification methods.

Importantly, the STEM-PCR strategies demonstrated here allowed us to develop systems that enable bisulfite-free methylation detection using genomic DNA or ctDNA. As proof of principle, working on patient samples from the clinic, we demonstrated the specific methylation detection of the *SEPTIN9* gene, paving the way for clinical applications. We also demonstrated the potential of the method to detect co-methylation in challenging real clinical samples, such as plasma, with mechanisms that have the added advantage of minimizing the impact of DNA shearing, which often occurs during centrifugation steps, needle shearing or in circulating DNA sequences in biological samples, which can introduce bias as to the cut sites[30].

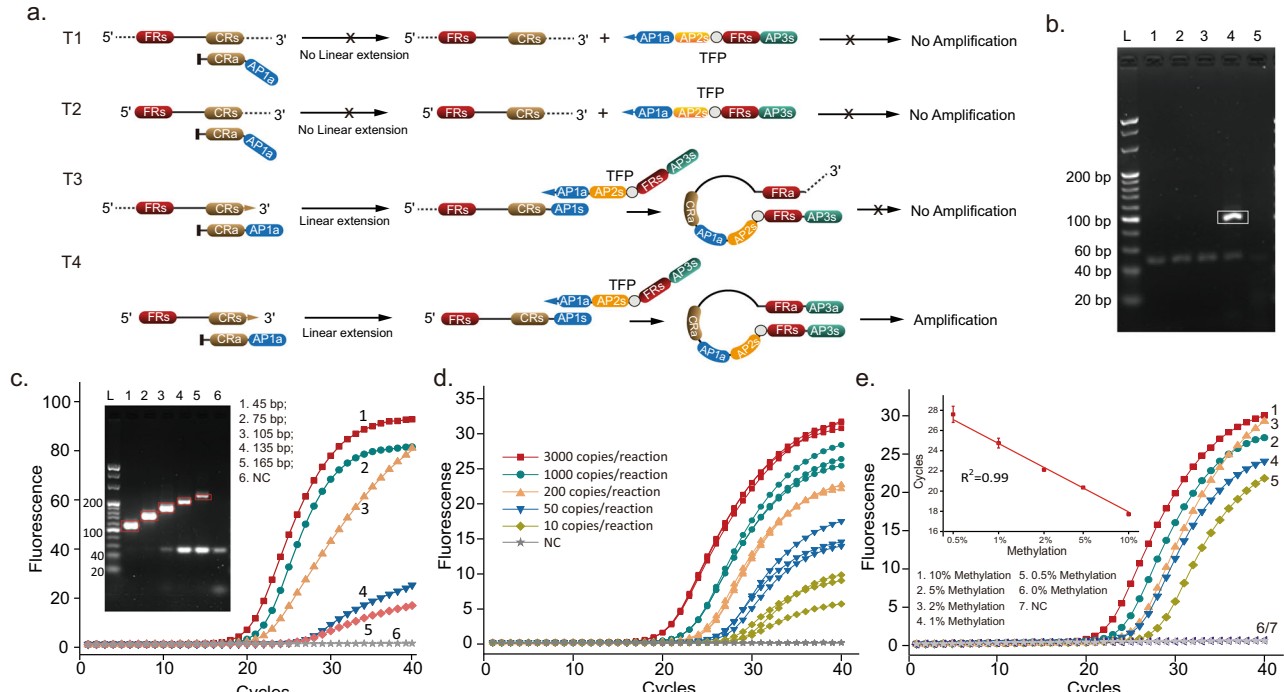

**Fig. 6 | STEM-PCR for different arbitrary breaks. a** Mechanisms for three different arbitrary breaks with different distances to 3′ and 5′ ends. T1 indicates no breaks at two ends. T2 and T3 refer to fragments with a break only at 5′ end and 3′ end, respectively. T4 is the fragment with two specific ends; **b** Methylated *SEPTIN9* sequences were treated with two different MDREs to generate specific 3′ and 5′ end, respectively. GlaI for 5′ end, while LpnPI for 3′ end. (1) no MDREs; (2) GlaI only; (3) LpnPI only; (4) GlaI and LpnPI; (5) ddH₂O as negative control. The band in a white box indicates the expected amplicon (100 bp); each experiment was repeated independently at least three times with similar results. **c** STEM-PCR amplification results of five artificial methylated fragments with different P1 lengths using the same primer set, using GlaI. (1) 45 bp (red); (2) 75 bp (green); (3) 105 bp (yellow); (4) 135 bp (blue); (5) 165 bp (magenta); (6) ddH₂O as negative control (gray).

Inset—Agarose gel of the amplification products. Target bands within white boxes correspond to different lengths of P1. L, 20 bp ladder; each experiment was repeated independently at least three times with similar results. **d** Sensitivity of STEM-PCR using GlaI: 3000 copies/reaction – red squares, 1000 copies/reaction – green disks, 200 copies/reaction – yellow triangles, 50 copies/reaction – blue inverted triangles, 10 copies/reaction – green lozenges, NC negative control – gray star; **e** Detection of different ratios of GlaI-cut methylated fragmented DNA in a background of 10 ng GlaI-cut unmethylated DNA. (10% - red squares, 5% - green disks, 2% - yellow triangles, 1% blue inverted triangles, 0.5% - green lozenges, 0% - gray stars, negative control – blue triangles). Data were the average of three replicates and error bars represent the standard deviation. The data were fitted with linear regression ($R^2 = 0.99$). Source data are provided as a Source Data file.

However, it should also be noted that the capability of STEM-PCR for location-specific methylation detection depends on the specificity of the digestion by MDREs, which can be limited for some methylated CpG and does not allow the differentiation of single mutations in co-methylation analysis. Furthermore, for each deletion, a specific set of PNA-modified blockers or specific TFP and FRs need to be designed.

STEM-PCR could conceivably advance DNA methylation analysis, improving the accuracy of detection and reducing complexity and time requirements, with a broad range of applications in epigenetic research, including the detection of hypomethylation, copy number variation, and microRNAs, for example. A wider application in biomarker discovery will require large-scale clinical studies to establish the method's performance in identifying new targets, and make use of its sensitivity and specificity for determining the potential modifications at an early stage of disease, where the changes at nucleic acid level is usually small. Described here for methylation detection, STEM-PCR also has potential for use in large-scale cancer screening, by, in future, integrating with high throughput automatic nucleic acid detection system, as a designed assay towards a set of targets, into an automated workflow (the technique is compatible with conventional PCR infrastructure).

## Methods
### Ethical statement
The research complies with all relevant ethical regulations, as well as all requirements from China's Ministry of Science and Technology. All the tissue samples were obtained from patients undergoing surgery at

Renji Hospital (Shanghai, China), with written informed consent having been obtained prior to surgery. No compensation was provided to participants. Protocols were approved by the Human Research Ethics Committee of Renji Hospital (approval RGH 09/04).

### Oligonucleotide and DNAs
The Jurkat genomic DNA, methylated Jurkat genomic DNA, and CpG methyltransferase (M.SssI) were obtained from Thermo Fisher Scientific. The MspJI and FspEI were purchased from NEW England Biolabs (Shanghai, China), while GlaI was from SibEnzyme (Russia). All the oligonucleotides and Taqman probes were obtained from Shanghai Sangon Biological Corporation (Shanghai, China) - Supplementary Tables 1–7. The online IDT oligoAnalyzer tools (https://eu.idtdna.com/pages/tools/oligoanalyzer) were used to avoid the formation of hairpin structures in TFP and dimers, in order to obtain a high amplification efficiency. To design FRs/a to improve self-folding efficiency, they were 15–20 nt in length, with 50-70% GC content. Champagne TaqTMDNA polymerase and Phanta UC Super-Fidelity DNA Polymerase for Library Amplification were purchased from Vazyme Biotech Co. Ltd.

The target sequence was first inserted into vector PuC57 and 300 ng of the constructed plasmid was incubated at 37 °C for 1 h with M.SssI to methylate the cytosine residues in the double-stranded dinucleotide CG sequence before extraction using QIAamp DNA Mini Kit (QIAGEN). The copy number of methylated and unmethylated DNA were quantified using Crystal digital PCR with ten times serial dilution.

## Tissue genomic DNA and cfDNA extraction

The genomic DNA was extracted according to the instructions of the QIAamp Fast DNA Tissue Kit, and the purity and quantification of the obtained DNA were assessed by absorbance using a Nanodrop spectrophotometer (Thermo Fisher). Only extracted DNA with absorbance ratios of A260/A230 > 1.8 and A260/A280 > 1.8 was used. All extracted DNA was stored at −20 °C for less than 2 months before use.

Spin column-based cfDNA extraction (QIAamp Circulating Nucleic Acid Kit, lot No.55114) was performed according to the recommended protocol. The volume of human plasma varied from 1 to 2 mL. 60 µl of elution buffer was applied in the final elution step. The purity and quantification of extracted cfDNA was tested using Nanodrop. The fragmented size was evaluated using Qsep1.

The FFPE blocks were obtained from 2012/02/07–2012/10/25 with written informed patient consent and were stored in the dark at room temperature. The study was approved by the Ethics Committee of Renji Hospital.

## Cut helper-mediated location-specific cut

Using MDREs requires target sequence specific-cut helper to hybridize to a methylated template to form a partial hemimethylated complex, resulting in template strand breaks at fixed distances from target methylated cytosine[16]. For any single specific CpG site, both of sense and antisense strands can be used to avoid the potential of forming a secondary structure. The cut helper is about 20 bp ($T_m$ = 65–70 °C) and encompasses the target CpG site at a position close to the 3′ end.

Target sequence specific-cut helper (50–100 nM) were incubated with genomic DNA in the reaction buffers recommended for the different MDREs at 95 °C for 5 min for melting, 60 °C for 20 min to form the hemicomplexes, and 37 °C for 30 min for the digestion, within a total volume at 10 µl. The MDREs digestion was then inactivated by heating to 90 °C for 10 min. The digested DNA can be stored at −20 °C until used, in our case, always for less than 2 weeks.

## Real-time amplification for STEM-PCR

The real-time amplification process of STEM-PCR was performed on a LightCycler instrument 480 II (ROCHE) in 20 µl, containing 10 nM TFP, 0.2 µM TSP, 0.2 µM AP, and 0.1 µM Taqman probe, 0.3 mM of dNTP (containing dUTP), 1×PCR buffer, 1 U Champagne Taq DNA Polymerase, 1 U UDG, and 10 µl of the sample (digested or not depending on the experiment). The mixture was heated at 95 °C for 5 min, 10 cycles of 95 °C for 10 s, 66 °C for 90 s, and followed by a second stage of 40 cycles of 95 °C for 10 s, 65 °C for 30 s. The detection was carried out during the annealing step by monitoring the fluorescence signal. Microsoft Excel 365 was used to analyse the curves, and results were plotted with Origin (OriginLabs, v2016).

## STEM-PCR for co-methylation detection

We obtained whole genomic DNA from HeLa cells (at ca. 50 ng/µL, BioChain Institute Inc., cells grown from stock from ATCC, catalog number 30-2003), and fragmented it using the Covaris M220 (75 W and 510 s). The main fragment size was 150 bp (established using Qsep1 capillary electrophoretic system, BioOptic Inc). MDRE-treated product was added to the reaction mix within the total 15 µl volume, containing 15 nM BP, 20 nM TFP, 1 X Champagne Taq™ Buffer (Mg²⁺ plus), 0.3 mM dNTP mix, and 0.05 U Champagne Taq™ DNA Polymerase. The reaction was performed on a ProFlex™ PCR instrument: 5 min at 95 °C, followed by 15 cycles of 10 s at 95 °C and 90 s at 66 °C.

Real-time amplification was carried out with a total 20 µl volume containing 0.2 µM AP2s, 0.2 µM AP3s, 0.2 µM Taqman probes, 1 X Champagne Taq™ Buffer, 0.3 mM dNTP (including dUTP), 0.5 U Champagne Taq™ DNA Polymerase, 1 U UDG and the mixture of the linear extension reaction. The suitable cycling condition was 37 °C for 5 min, 95 °C for 5 min, 10 cycles of 95 °C for 10 s and 66 °C for 90 s, followed by 40 cycles of 10 s at 95 °C and 30 s at 65 °C. The fluorescent signal was measured during the annealing step.

## Digital PCR

Crystal digital PCR reactions were prepared using Perfecta multiplex qPCR ToughMix (Quanta Biosciences, USA) and a 4 µl template. About 40 nM of reference dye was added to allow adequate imaging of all droplets for analysis. To prepare Sapphire chips, 20 µl of PCR mix were pipetted into the inlet ports before the pressure-permeable caps (Stilla Technologies) are positioned. The cycling conditions of the ddPCR were: 95 °C for 10 min, followed by the two-stage process: 10 cycles of 95 °C for 10 s, 66 °C for 90 s, then 40 cycles of 95 °C for 10 s, 65 °C for 30 s.

Image acquisition was performed with Naica Prism 4 reader with the following exposure times: blue channel: 100 ms; green channel: 50 ms; red channel 50 ms. The total droplet enumeration and droplet quality control were calculated using reference dye (blue channel). Extracted fluorescence values for each droplet were analysed using Stilla Crystal Miner v2.4.0.3 automatically.

## Bisulfite conversion-sequencing

One microgram of genomic DNA extracted from each of the 20 tissue samples were converted following the instruction of EpiTect Fast DNA Bisulfite Kit (QIAGEN). The converted DNA was amplified using the primers (Supplementary Table 1) before sequencing. The amplification was as follows: 0.4 µM primers, 1x Uc Buffer for Library Amplification, 0.2 mM each dNTP, 1U Phanta Uc Super-Fidelity DNA polymerase. The reactions were carried out on the ProFlex™ 3 ×32-well PCR System Applied Biosystems™ with the following program: initial denaturation at 95 °C for 3 min; 35 cycles of 94 °C for 30 s, 60 °C for 30 s, 72 °C for 60 s; then incubation at 72 °C for 7 min. The amplicons were analysed using gel electrophoresis, the band was cut and purified using QIAquick Gel Extraction Kit before the quantification with Qubit 4 Fluorometer. the PCR products were tested using Sanger sequencing. The inconsistent results were diluted 10,000 times and served as a template for nest methylated specific-PCR to generate the methylated specific-PCR product for a second round of sequencing. Sequencing data were analysed using Chromas (Technelcium Pty, v2.6.6).

Plasma cfDNA was extracted by QIAamp® Circulating Nucleic Acid Kit (Qiagen, 55114) according to the manufacturer's protocol. DNA was subjected to the bisulfite conversion step using EZ DNA Methylation-Lightning™ Kit (Zymo Research, D5030) according to the manufacturer's protocol.

Targeted genome methylation analysis was conducted using the OPERA Mars® Universal Library Prep Kit (Apogenomics, APG-62001), OPERA® Index Adapter (Apogenomics, APG-23005A), and OPERA® Index Primer (Apogenomics, APG-23009A) according to the manufacturer's protocol. Briefly, the procedures were divided into three main steps: (i) multi-cycle linear amplification using a single-primer panel, (ii) ligation between amplified ssDNA product and single strand UMI adapter, which contains index sequence, (iii) indexing PCR for amplifying the ligated product with Index Primer. These target-enriched libraries were further amplified with P5 and P7 primers and purified for sequencing on the NovaSeq 6000 System (Illumina Inc).

Targeted bisulfite conversion sequence data were first trimmed by cutadapt 1.18. The reads were mapped to targets around 600-700 bp reference sequence with bismark v0.19.1, and transferred to remove duplication reads with fgbio 1.0.0 following standard instructions. The deduplicated reads were mapped to reference and the co-methylated reads were counted by manual inspection in the integrative genomics viewer (IGV v2.8.9). Reads containing more than 4 methylated CpG site was classified as co-methylated.

## Statistics and reproducibility

No statistical method was used to predetermine the sample size. No data were excluded from the analyses. The experiments were not randomized and the Investigators were not blinded to allocation during experiments and outcome assessment; however, the benchmark and STEM-PCR experiments were performed in parallel.

## Reporting summary

Further information on research design is available in the Nature Portfolio Reporting Summary linked to this article.

## Data availability

The sequencing data generated in this study have been deposited in the Sequence Read Archive (SRA) of the National Center for Biotechnology Information (NCBI) under accession number "PRJNA930891". The data generated in this study have been deposited in The University of Glasgow repository Enlighten at https://doi.org/10.5525/gla.researchdata.1298. Source data are also provided with this paper. Source data are provided with this paper.

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

## Acknowledgements

We are grateful for access to FFPE samples provided by Professor Ming Zhong (Renji Hospital). This work was supported by the SJTU STAR Interdisciplinary Important Program (20190201, to H.G.), Innovation Research Plan from the Shanghai Municipal Education Commission (ZXWF082101/056, to G.X.), Natural Science Foundation of Shanghai Exploratory Project (19ZR1476000 to G.X.), key projects of special development funds for Shanghai Zhangjiang National Innovation Demonstration Zone (201905-XH-CHJ-H25-201 to H.G.), as well as the UK Research and Innovation through the Medical Research Council (MR/V035401/1 to J.M.C.) and the Biotechnology and Biological Sciences Research Council (BB/T012528/1 to J.M.C.).

## Author contributions

G.X. conceived the original idea, J.Q. and H.Y. designed the experiment supervised by G.X., H.G., and H.X.; H.Y., J.Q., L.Z., and W.R. performed the experiments; G.X., H.Y., J.Q., J.R., and J.M.C. analysed data; G.X., J.R., and H.X. wrote the manuscript draft. All authors revised the text.

## Competing interests

The authors declare no competing interests.
