## [Peer Review File · Nature Communications]

REVIEWER COMMENTS

Reviewer #1 (Remarks to the Author):

In the paper, the authors present a new PCR-based method for identifying site-specific mutations and modifications. The paper is well written and the scientific background behind the method is sound. The authors apply clever PCR-based solutions to distinguish between presence and absence of specific artifacts. In general, most experiments seem to be performed on synthetic input material. Additional tests of the method on real-life samples could further help assess the robustness and sensitivity of the approach in a less artificial setting.

Minor points:

- I think up to Figure 1 TSP has not yet been explained. The abbreviation is used in that figure but has not been addressed before.
- In figure S5, it is not clear why the band in the first lane (using the 1st setup having both CpG sites at the recognition sequence) indicates shorter products than in the other three setups.
- In the section starting at line 267 it is unclear whether synthetic or genomic input material was used.
- In general, how robust is the method in coping with fragmented DNA (e.g. originating from liquid biopsies). For example in the PNA modification-based method, if in a wild-type fragment the region to which the PNA modified oligo needs to bind is partly absent, there's a high chance that elongation and amplification will be successful designating that fragment incorrectly as mutant. An experiment showing the impact of the level of fragmentation on the method would be interesting.
- When comparing figures 3a and 4b it's clear that the amplification efficiency of the RE-based approach is much better than the PNA modification-based approach when using a similar number of copies per reaction. Why is that? Could it be that the PNA modified oligo (targeting the wild type) is incorrectly binding the mutated site as well resulting in decreased/under-estimated detection of the latter? An experiment on mutated in wt-background dilutions to assess the methods sensitivity in such cases could help clarify this. The same holds true when applying the method for co-methylation detection (cfr figure 6 – what if the break occurs in either the FRs or CRs region)?
- What is the nature of the extension blocker in the TFP in figures 1 and 1S?
- I'm missing information in the materials and methods section of the design parameters (length, GC content, annealing temperature, ...) for the different oligos (e.g. TFP) and the conditions to which for example the FRs/a must abide to enable efficient hairpin formation.

- Possible typos:

1. line 97 : ..., including for multiple sites at a time, ...
2. line 363 : ... could lead to step changes in DNA methylation analysis, ...
3. line 367 : ... screening at community clinics by integrating ...
4. line 368 : ... nucleic acid detection systems, as it is fully ...
5. line 396 : ..., both of sense and antisense strands can be used ...
6. line 422 : ..., with the fluorescent signal was obtained during ...
7. line 429 : ... are positioned onto the loaded ports.

Reviewer #2 (Remarks to the Author):

- What are the noteworthy results?

==> NO - authors should submit to a methods journal like Biotechniques etc...

- Will the work be of significance to the field and related fields? How does it compare to the established literature? If the work is not original, please provide relevant references.

==> another PCR based assay as many other,

- Does the work support the conclusions and claims, or is additional evidence needed?

==> partially – the claimed “broad application options” are not really shown

- Are there any flaws in the data analysis, interpretation and conclusions? - Do these prohibit publication or require revision? ==> NO

- Is the methodology sound? Does the work meet the expected standards in your field?

==> YES

- Is there enough detail provided in the methods for the work to be reproduced?

==> YES

PS: I assume the word "detection" is missing at the end of the TITLE

Reviewer #3 (Remarks to the Author):

In the manuscript, the authors attempted to report the sequence terminus dependent PCR for site-specific mutation and modification, named Specific Terminal Mediated Polymerase Chain Reaction (STEM-PCR). The authors have demonstrated that the methods can be applied both for methylation and single base mutations analysis in clinical applications, using a bisulfite-free process with a higher sensitivity and specificity than standard BS-PCR sequencing. This research has certain value on clinical applications. However, there are some concerns on the manuscript.

1. In Figure 3, methylated sites of SEPTIN9 and SFRP2 genes were detected to compare the performance of STEM-PCR. But only SEPTIN9 was mentioned in the abstract.

2. In line 347, there is the grammatical mistake.

Reviewer #4 (Remarks to the Author):

Comments for authors

The authors describe a well-conducted study illustrating their novel STEM-PCR technique that allows the detection of DNA modifications at specific sites, comparable to DNA sequencing techniques. This study is a nice overview of the functionality of STEM-PCR, and the authors have done efforts to perform elaborate experiments to show this functionality, as well as a comparison with the current gold standard.

Although the possibility to easily detect methylation without the necessity for a bisulfite conversion could definitely further advance the epigenetic (biomarker) research field, I am not yet convinced of the clinical applicability of this technique for methylation biomarker purposes. From an epigenetic biomarker point of view, this study raises some questions that should be answered/addressed more elaborately.

Major comments

1. STEM-PCR shows a high level of detection/high resolution which in itself is very good. However, it can be questioned whether this high resolution is that positive when it comes to biomarker research.

Especially when identifying or searching for potentially candidate markers, the identification of a small number of modified copies in a bulk of “normal” DNA could also lead to false positive candidate biomarkers that are then studied in subsequent validation studies without results. Is it envisioned that this technique is therefore only suitable for the validation, or the detection in clinical practice, of already validated and established biomarkers, and not for the identification of novel markers?

2. Most of the experiments in this study have been performed on DNA of high quality. However, for biomarker research the majority of researchers need to use archived material, FFPE tissue that sometimes is over 10 years old, and preserved or prepared using different protocols. How does STEM-PCR perform when using this sub-optimal DNA? Let alone, ctDNA or cfDNA from clinical liquid biopsy samples that have been frozen for some time. The technique had not been tested on these samples yet, even though for biomarker research researchers over the world depend on these low quality DNA samples. Before any statements on the use in biomarker research can be made, the technique should be tested on these samples (older FFPE tissue from archives).

3. Following comment 2, STEM-PCR has only been tested on clinical tissues, while most epigenetic biomarker assays aim to detect methylation in liquid biopsies as this is perceived to be more clinically useful for most applications. STEM-PCR has been tested on simulated ctDNA (fragmented) but this is not the same as ctDNA (or cfDNA) from clinical liquid biopsies that could contain even smaller fragments and are of lesser quality. So, before statements on clinical applications can be made, the technique should be tested on ctDNA or cfDNA from clinical liquid biopsies, preferably liquid biopsies from archives or biobanks and not only on simulated ctDNA.

4. The authors indicate that STEM-PCR would be suitable for a broad range of applications in epigenetic research, which is indeed probably the case. It is also stated that this technique has potential for large-scale cancer screening, and in a clinical biomarker testing setting. This latter application is more challenging in my opinion, given the number of steps and modifications that need to be planned in the approach and the high number of custom/tailored components that are necessary to complete STEM-PCR. The technique appears to be quite complex due to these number of steps and tailored components, and therefore this might be too difficult to implement in clinical practice. At the moment, established high throughput detection systems are not always present in clinical institutions and some institutions lack the expertise to introduce these systems. STEM-PCR in itself can probably not be implemented at these sites, and even if integrated in established systems or in a commercially available kit, specific training will probably be necessary as there are many steps in the technique where problems or errors could occur. In addition, the technique is currently far from translation into a commercial kit, as it will need a lot of validation work on clinical samples of all sorts (see also comment 2). The message of the potential clinical application of this technique should therefore be toned down in the manuscript in my opinion.

5. Following comment 2 and 3, it can be questioned whether STEM-PCR is not much more clinically valuable to detect mutations, instead of methylation. In the case of known mutations, will it not be much more easier to design a commercial kit specifically targeted at that known mutation? Easier then designing a commercial kit for methylation detection?

6. No additional information is given on how to design or select the custom/tailored components that are necessary for STEM-PCR. Are tools to help the user available for this? Also for this part, specific training seems necessary before users can work with STEM-PCR.

7. Methylated SEPT9 is used as an example to show the potential of STEM-PCR in biomarker research. For this marker, a commercial assay has long been available (EpiProColon). For a clinically useful comparison with STEM-PCR you would therefore expect a comparison with this commercial assay, why was this assay not used?

8. STEM-PCR is currently only compared to techniques that are similar to STEM-PCR. It would be interesting to see the comparison with other available (commercial) kits, to compare performance.

9. Figure 6 in the supplemental files shows the results of Sanger sequencing, depicting an almost perfect picture of sequencing results. Were these results actually derived from clinical samples as suggested in the text? In our hands, we have never seen such a perfect sequence chromatogram in clinical tissue samples as there always is a mixture of normal unmethylated and tumour methylated tissue. Using this approach, you could miss the methylated allele as there is an abundance of unmethylated alleles. To overcome this, bisulfite sequencing of cloned alleles could be used as this would provide a more quantitative and more sensitive result. This approach could in fact lead to the picture as shown in Figure S6, but it has not been described in the materials and methods that this approach was used and that cloned alleles were sequenced. What is the approach used here, which results are shown in S6? If BS-seq is not performed on clones, the technique is much less specific, STEM-PCR should therefore be compared to BS-seq on cloned alleles to better illustrate the comparison of the two techniques.

10. Line 446: Inconsistent results were diluted 10,000 times and served as template for nest MSP to generate the methylated specific-PCR product for a second round of sequencing.

```
gttgtagggatgggtttgtaattgttgCGttttaggagaagtCGggttggttaaagaggaagagttCGCGCGtCGagtttagtagtagtg
CGaggCGaggaagagtagtagtagCGagttagggttttagtagatCGtgggCGCGCGatttCGaggggtagagggagCGgagtCGg
ggaagggCGaggCGgtCGgagttCGagtttgtttCGggttCGtttttttCGttgggtgCGattCGgggtttCGaaaagttgtagtC
GgCGgttggggCGCG
```

Primers are not properly marked in the table, and it is not specified which primers are used for which technique, but it can be assumed that the blue/bold primers are for BS and the yellow/italics primers for MSP. However, if you would dilute the blue product 10,000 times, the yellow product cannot be amplified. Is the description given in the manuscript correct? Have primers been depicted correctly? This should be checked.

BS-PCR

SFRP2PF1 GGAGAAGTAGGGTTGGTTAAAGAG

SFRP2PR1 ACTCCCCTCCCTCTACCC

Rev comp gggtagagggaggggagt

SFRP2PF2 GTTTAGGTAGTAGTGCAGGC

SFRP2PR2 CTACCAACTTTTCGAAACCCC

Rev comp ggggtttcgaaaagttgtag

Minor comments

1. Figure 1a, number 2, extension blocker is depicted below the figure, is this correct?
2. Numerous abbreviations are being used in the article and even in the figures, making it difficult to read. A list of abbreviations would be helpful.
3. In line 217 the authors reference article [19] which describes reduced representation bisulfite sequencing libraries for genome-scale DNA methylation profiling; a technique for whole genome sequencing. In the Materials and Methods line 436, the authors refer to Sanger sequencing which is a different technique. Reference 19 does not refer to the standard Sanger sequencing as is implied in the text, is this reference correct?
4. No references are used when explaining the MSP methods, is this correct? Would be better to add some references.
5. Figure 5, the red dotted box is not visible in the figure.
6. Figure 1 from the supplemental file should be moved to the main text.

Reviewer #5 (Remarks to the Author):

The manuscript by Xu et al entitled "Sequence terminus dependent PCR for site-specific mutation and modification" describes a new approach for detection of alterations in DNA sequence or DNA methylation. This method can assess single nucleotide changes (mutations) via a blocking oligonucleotide, or methylation sensitive restriction endonucleases to allow detection of methylation changes at a single CpG site.

The manuscript describes the approach well, and while a complicated approach to detect such changes, carefully demonstrates utility primarily on artificial templates mixed in unmodified DNA to demonstrate sensitivity of this approach. The primary advantage of this method for methylation detection is that unlike other restriction endonuclease methods which rely primarily on the absence of cleavage to detect methylation, this method only detects methylation when methylated sequences are cleaved (and thus the incomplete cleavage of previous methods generating a false positive is eliminated). It is possible that star activity of the RE could produce a false positive and might be discussed, but may be minimal for these endonucleases.

For mutational detection, a blocking oligo prevents the necessary structure for amplification.

Detailed figures in the manuscript show the potential use of such an approach, which is described as providing single nucleotide resolution of DNA methylation or mutation without sequencing, and doing so with greater sensitivity than sequencing.

While these advantages are nicely described and shown, the limitations of STEM-PCR are not mentioned.

First, the limitation for methylation detection to examination of single CpG sites requires these lie in a methylation sensitive RE recognition sequence. Other sites would not be amenable to examination. In addition, the requirement to design a specific assay for each CpG site is also complex, rather than the ability to determine DNA methylation status at multiple sites within a sequenced region. In this latter consideration, the detection of co-methylation is demonstrated in figures 5 and 6, we an assay using two methylation sensitive RE recognized sites to detect concordant methylation at these two sites, but this would not discriminate unmethylated from sequences methylated at either of the two sites, which would I believe require use separate digestion of template with only one RE to detect partial methylation, greatly complicated the analysis. In contrast, bisulfite sequencing or alternative

modification sequencing approaches recently developed provide individual allele patterns of the 4 possible states, as well as the status of adjacent CpG sites.

The sensitivity of this approach is shown, and in supplemental data reported to be 20 times more sensitive than Heavy Methyl PCR, which is stated in the manuscript to thus be 20 fold more sensitive than the state of the art. Heavy Methyl is a difficult approach only developed for commercial use for IP issues, and is not the most sensitive approach for methylation detection. As shown in the data in figure 3, while STEM-PCR was more sensitive than sequencing (expected), the confirmation Methylation Specific PCR then assessed for this site methylation by sequence verification was equal to STEM-PCR. The reported sensitivity of 0.1% methylation in the manuscript was reported in the original MSP manuscript, with realtime MSP showing greater sensitivity. Other newer approaches have also shown much greater sensitivity than 0.1% and reach single molecule detection. Figure 3 does show reasonably sensitive detection, but doesn't provide any confirmation that this is specific (ie no control patients were included). This is important to assess, since the detection of single site methylation (which is the focus of STEM_PCR) may not provide the level of specificity needed for this clinical application.

The mutational detection of STEM-PCR is demonstrated for EGFR detection of the common L858R mutation in figure 4. Again, showing detection down to 30 copies, this is a sensitive detection approach and is shown to be specific. The limitations of this analysis to just a single mutation in EGFR is not discussed, and such an approach would require a specific assay for each of the known point mutations in this gene, and would not readily I believe detect the deletion mutations also common (this may not be true, if the PNA would cross a deletion site, but was not discussed and would also require different PNAs for each potential deletion).

We thank the reviewers for their comments and have used their feedback to improve the paper making clarifications where necessary. The new version of the paper, as requested, also includes additional experiments, including new clinical data, to support the research.

We have answered all comments in a point-by-point reply (comments/questions are shown in black and our **responses in red**, for convenience). We believe that these corrections have greatly improved the quality of the manuscript and we are grateful for the reviewers' attention in their evaluation of the manuscript.

Reviewer #1

In the paper, the authors present a new PCR-based method for identifying site-specific mutations and modifications. The paper is well written and the scientific background behind the method is sound. The authors apply clever PCR-based solutions to distinguish between presence and absence of specific artifacts. In general, most experiments seem to be performed on synthetic input material. Additional tests of the method on real-life samples could further help assess the robustness and sensitivity of the approach in a less artificial setting.

R1.0: We thank the Reviewer for the positive views on the capabilities of our new technique. We would like to emphasize that we do provide tests on "real-life" samples, including extracted genomic DNA from patients in a similar platform as would be performed in routine processes at the hospital. In response to Reviewer 4, we have also now included work on cfDNA from blood plasma from 14 patients (R4.2).

Minor points:

Q1.1 I think up to Figure 1 TSP has not yet been explained. The abbreviation is used in that figure but has not been addressed before.

R1.1: We have added the definition of TSP inside Figure 1 and re-drawn the Figure 1 as suggested by both Reviewer 1 and Reviewer 4 (Q4.16).

Q1.2 In Figure S5, it is not clear why the band in the first lane (using the 1st setup having both CpG sites at the recognition sequence) is indicates shorter products than in the other three setups.

R1.2: Each TFP used in the experiment of four different cleavage scenarios were different (Table S2). We have now added an additional note of clarification in the caption as follows:

"It should be noted that the difference in sizes between lanes 1 and (2-4) arises from the fact that the TFP sequence for each scenario was different, leading to different size fragments (i.e. 2-4 also are different)."

Q1.3 In the section starting at line 267 it is unclear whether synthetic or genomic input material was used.

R1.3: We apologise for the misunderstanding. We extracted whole genomic DNA (at ca. 50 ng/μL), and fragmented it using Covaris M220 with the power and incubation time at 75 W and 510 seconds, respectively. The main fragmented size was 150 bp (established using Qsep1 capillary electrophoretic system, BioOptic Inc). We have changed the text accordingly:

To demonstrate this, human genomic DNA was extracted from cultured cells and fragmented to a size of ca. 150 bp (Figure S10) as a model for methylated ctDNA.

We also added details to the Methods Section on co-methylation, accordingly:

We obtained whole genomic DNA from HeLa cells (at ca. 50 ng/μL, BioChain Institute Inc.), and fragmented it using the Covaris M220 (75 W and 510 seconds). The main fragment size was 150 bp (established using Qsep1 capillary electrophoretic system, BioOptic Inc).

Q1.4 In general, how robust is the method in coping with fragmented DNA (e.g. originating from liquid biopsies). For example in the PNA modification-based method, if in a wild-type fragment the region to which the PNA modified oligo needs to bind is partly absent, there's a high chance that elongation and amplification will be successful designating that fragment incorrectly as mutant. An experiment showing the impact of the level of fragmentation on the method would be interesting.

R1.4: Theoretically, there are mainly two different states for fragmented DNA, as follows:

1. In the wild-type fragment, the region to which the PNA modified oligo needs to bind is partly absent. This leads to the absence of the PNA modified oligo in the reaction, allowing the extension of TFP, which then stops at the 5' end of the sequence. After self-folding, the overhung part prohibits the extension of FRa, thus stopping the formation of P2 and the exponential amplification. We provide a new Figure S9a to visualize these scenarios.
2. When the 5' end of the fragmented wild-type is at the same position as the 5' end of PNA modified oligo, it leads to amplification of STEM-PCR and a false-positive.

Following the recommendation of this Reviewer, we have performed additional experiments to evaluate the impact of these behaviours on the performance of STEM-PCR. Figure S8b and Figure S9 show that the level at which these interferences arise is low (e.g. when we used 3000 copies wild-type background, there was no amplification).

We have modified the text to discuss the two new Figures, included here for ease of reference.

*“The impact of the level of template fragmentation on STEM-PCR was studied in Figure S9. When the truncated position site at the 5' end of FRs was partly absent from the PNA blocker, this prevented the hybridization of PNA, which in turn led to the extension of TFP, generating a short tail overhang after self-folding, and preventing amplification (identical to that for untruncated wild type sequences).”
[.] “The specificity was as low as 1% methylated mutated template detected in a background of 3000 copies of wild-type DNA (Figure S9b)”*

Figure S9. Impact of the level of fragmentation of template on STEM-PCR. (a). Different scenarios of fragmentation that affected the outcome of STEM-PCR. 1. Truncated sites of fragmented DNA were distant from the 5' end of FRs; 2. The truncated sites of fragmented DNA lay within FRs region. (b). The sensitivity of PNA mediated STEM-PCR for L858R mutation detection with 3000 copies as background. The limit of detection was below 1%.

Figure S12:

“The impact of the level of fragmentation on co-methylation detection was also studied using different artificial templates. The results of different lengths of FRc (Figure S12a) indicated that the truncated position at the 5' end of FRs prevented STEM-PCR, even with only single one extra base. We hypothesise that this may have been due to the overhung bases not only preventing the extension of FRA after self-folding, but also leading to mismatch of AP2s. When the truncated position sites were located within FRs regions, the amplification efficiency decreased with the decrease in generation of FRA. For example, the efficiency was ca 10 times lower when 2 and 3 bases were deleted from the standard FRs. A similar study was carried out to explore effect of CRs (Figure S12b), showing again a decrease in efficiency as the length of CRs decreased. One base missed from the CRs leads to a five-fold reduced efficiency, whilst more bases missed prevented amplification completely.”

Figure S12. The impact of the level of fragmentation on co-methylation detection using the SEPTIN 9 amplification system. Different artificial oligos were synthesized and severed as template (ca. 20 copies/reaction). (a) The oligos had the same CRs region but different FRs. The base overhung at the 5' end of templates inhibits the amplification, even with only 1 base change. The amplification efficiency decreased as the FRs sizes decreased. (b) The oligos had the same FRs region but different CRs. The amplification efficiency decreased as the length of CRs decreased.

Q1.5 When comparing figures 3a and 4b it's clear that the amplification efficiency of the RE-based approach is much better than the PNA modification-based approach when using a similar number of copies per reaction. Why is that? Could it be that the PNA modified oligo (targeting the wild type) is incorrectly binding the mutated site as well resulting in decreased/under-estimated detection of the latter? An experiment on mutated in wt-background dilutions to assess the methods sensitivity in such cases could help clarify this. The same holds true when applying the method for co-methylation detection (cfr figure 6 – what if the break occurs in either the FRs or CRs region)?

R1.5: We thank the Reviewer for this detailed analysis. Indeed, REs digestion is more efficient (usually >99%) (Electrophoresis. 2007,28(10):1514-7.)(reference 25) than the hybridization of the PNA blocker (ca. 90–97% (Gen. Chrom Cancer. 2001,30:57–63) (reference 26), indicating 3%-10% loss of target sequence. This effect is compounded by the fact that this lower efficiency is amplified at the stage of the synthesis of the full hairpin structure. We also note that the effect of truncation (discussed in Q1.4) will have no impact on RE digestions (Figure S9a).

We added this discussion and a following statement to explain the specificity of PNA-based approach for mutation detection with 3,000 copies wild-type background have been included, accordingly.

“The specificity was as low as 1% methylated mutated template detected in a background of 3000 copies of wild-type DNA (Figure S9b).”

Q1.6 What is the nature of the extension blocker in the TFP in figures 1 and 1S?

R1.6: The extension blocker used in our study is polyethylene glycol (PEG18) similar to 6 nt in length. The long distance between CRa and FRs can effectively inhibit the extension by Taq polymerase, to improve the linear amplification efficiency after the P3 product is generated. (see J. Am. Chem. Soc. 1992,114: 8768-8772) (reference 16).

We have added this detail in the caption of Figure 1 and in the tables of primers S1.

Q1.7 I'm missing information in the materials and methods section of the design parameters (length, GC content, annealing temperature, ...) for the different oligos (e.g. TFP) and the conditions to which for example the FRs/a must abide to enable efficient hairpin formation.

R1.7: We agree that the information on the primers was confusing, since it was presented in a single table in supplementary material. We have now split this information in different tables. Although, it is challenging to provide a set of generic rules to design complementary sequences based on the mechanisms provided in the Figures of the main text, we have added the software used to check on sequences (from IDT) in the Methods section. Figure 1 further contains quantitative information on the primers used. We would also recommend to design FRs/a with a view to improve self-folding efficiency, towards a length of 15-20 nt, with 50-70% of GC.

We have now added this detail as short note to the Methods Section. We would hope that this information together with the sequences provided would serve as a useful starting point for researchers willing to design their own assays.

Q1.8 Possible typos:

1. line 97 : ..., including for multiple sites at a time, ...
2. line 363 : ... could lead to step changes in DNA methylation analysis, ...
3. line 367 : ... screening at community clinics by integrating ...
4. line 368 : ... nucleic acid detection systems, as it is fully ...
5. line 396 : ..., both of sense and antisense strands can be used ...
6. line 422 : ..., with the fluorescent signal was obtained during ...
7. line 429 : ... are positioned onto the loaded ports.

R1.8 We thank the reviewers for highlighting these. Often, these issues arose from grammatical errors in sentences that were long. We have modified the text where appropriate, to shorten the statements and improve the overall understanding. In general, all of the authors have also proof-read in detail the whole manuscript before resubmission, leading to other minor changes in grammar or syntax, not directly linked to substantial changes in meaning but to improve readability.

Reviewer #2

- *What are the noteworthy results?*

==> NO - authors should submit to a methods journal like *Biotechniques* etc...

- *Will the work be of significance to the field and related fields? How does it compare to the established literature? If the work is not original, please provide relevant references.*

==> another PCR based assay as many other,

- *Does the work support the conclusions and claims, or is additional evidence needed?*

==> partially – the claimed “broad application options” are not really shown

- *Are there any flaws in the data analysis, interpretation and conclusions? - Do these prohibit publication or require revision? ==> NO*

- *Is the methodology sound? Does the work meet the expected standards in your field?*

==> YES

- *Is there enough detail provided in the methods for the work to be reproduced?*

==> YES

PS: I assume the word "detection" is missing at the end of the TITLE

R2: We disagree with the statement that the work presented here is “another PCR-based assay as many other”. We show the ability to detect sequence modification, which currently requires either complex and impractical modifications (such as bi-sulfite enrichment) or are carried out using expensive next generation DNA sequencing platforms.

We also would like to emphasize that the capabilities presented here have applications in the detection of nucleic acid modifications (as demonstrated), which has the potential to impact broadly on clinical analysis and decision-making. As stated in the text, the new technique has the potential to be applied into the wide field of epigenetic research, as well as for cancer screening in clinics.

We are remiss of the typographical error in our title which arose during formatting. It indeed should read “Sequence terminus dependent PCR for site-specific mutation and modification detection”. We apologise for the issue and have modified this throughout the submission.

Reviewer #3

In the manuscript, the authors attempted to report the sequence terminus dependent PCR for site-specific mutation and modification, named Specific Terminal Mediated Polymerase Chain Reaction (STEM-PCR). The authors have demonstrated that the methods can be applied both for methylation and single base mutations analysis in clinical applications, using a bisulfite-free process with a higher sensitivity and specificity than standard BS-PCR sequencing. This research has certain value on clinical applications. However, there are some concerns on the manuscript.

Q3.1. In Figure3, methylated sites of SEPTIN9 and SFRP2 genes were detected to compare the performance of STEM-PCR. But only SEPTIN9 was mentioned in the abstract.

R3.1: We apologise for the error and thank the Reviewer for pointing out this important point. We have added SFRP2 in the abstract. We also noted that the mention of both genes was confusing in the introduction and we clarified this with new references there. We have also taken this opportunity to shrink the abstract to meet the editorial requirements from the journal.

Q3.2. In line 347, there is the grammatical mistake.

R3.2 We thank the reviewer for pointing this out. As mentioned in our response to R1.8, we have proof-read the manuscript again thoroughly. Here again, the difficulties appeared to be linked to overly lengthy sentences.

Reviewer #4 (Remarks to the Author):

Comments for authors

The authors describe a well-conducted study illustrating their novel STEM-PCR technique that allows the detection of DNA modifications at specific sites, comparable to DNA sequencing techniques. This study is a nice overview of the functionality of STEM-PCR, and the authors have done efforts to perform elaborate experiments to show this functionality, as well as a comparison with the current gold standard.

Although the possibility to easily detect methylation without the necessity for a bisulfite conversion could definitely further advance the epigenetic (biomarker) research field, I am not yet convinced of the clinical applicability of this technique for methylation biomarker purposes. From an epigenetic biomarker point of view, this study raises some questions that should be answered/addressed more elaborately.

Major comments

Q4.1. STEM-PCR shows a high level of detection/high resolution which in itself is very good. However, it can be questioned whether this high resolution is that positive when it comes to biomarker research. Especially when identifying or searching for potentially candidate markers, the identification of a small number of modified copies in a bulk of “normal” DNA could also lead to false positive candidate biomarkers that are then studied in subsequent validation studies without results. Is it envisioned that this technique is therefore only suitable for the validation, or the detection in clinical practice, of already validated and established biomarkers, and not for the identification of novel markers?

R4.1: We agree the reviewer’s comment that STEM-PCR may not be the most appropriate for biomarker identification in the clinic at this stage of its development, but is instead, as with all applications demonstrated in this manuscript, highly suitable for biomarker validation and clinical applications (e.g. once the modification is known to be of significance).

Notwithstanding this, we do believe that the method has potential for cancer biomarker analysis early in the identification process, when high sensitivity and specificity are necessary for selecting the potential modifications, at an early stage of the disease (where the changes at nucleic acid levels are usually smaller). We note that the demonstration of the performance of STEM-PCR in biomarker discovery will require large population-based clinical studies to establish the risks for false positives.

We have added a general note to avoid any misunderstanding in the potential for the technique in the Discussion Section.

“A wider application in biomarker discovery will require large-scale clinical studies to establish the method’s performance in identifying new targets, and make use of its high sensitivity and specificity for determining the potential modifications at an early stage of disease, where the changes at nucleic acid level is usually small”

Q4.2. Most of the experiments in this study have been performed on DNA of high quality. However, for biomarker research the majority of researchers need to use archived material, FFPE tissue that sometimes is over 10 years old, and preserved or prepared using different protocols. How does STEM-PCR perform when using this sub-optimal DNA? Let alone, ctDNA or cfDNA from clinical liquid biopsy samples that have been frozen for some time. The technique had not been tested on these samples yet, even though for biomarker research researchers over the world depend on these low quality DNA samples. Before any statements on the use in biomarker research can be made, the technique should be tested on these samples (older FFPE tissue from archives).

R4.2: Following Reviewer 4’s suggestion, we identified and procured 10 samples which were over 10 years old and conducted STEM-PCR (using the mechanism from Figure 1 for SEPTIN9) on DNA extracted using a commercial kit by TAOGEN Inc. (concentrations 10.5 - 29.5 ng/μl; 260/280 ratio 1.94 - 2.12).

Results (Supplementary Figure S7) demonstrated a clear differentiation between cancer types. Unfortunately, we were not able to perform BS-sequencing (which requires at least 30 ng of DNA) due to the limited quantity of extracted genomic material.

(c)	Pathology	STEM-PCR Ct (average)	Standard deviation
1	Chronic inflammation of the sigmoid mucosa, polypoid growth	34.1	4.2
2	Chronic inflammation of the sigmoid mucosa, polypoid growth	35.2	2.0
3	Rectal adenocarcinoma, size 4*3.5*3 CM	26.9	0.3
4	Rectal adenocarcinoma, size 2*1.5*1.2 CM	29.7	0.6
5	Chronic inflammation of the sigmoid mucosa	35.0	0.4
6	Focal squamous hyperplasia	33.6	3.1
7	Rectal adenocarcinoma, CK+, CK7-, CK19+,EMA+	28.7	0.4
8	Rectal adenocarcinoma, 7.5*5.5*1.5 CM	25.2	0.2
9	Chronic inflammation of the rectal mucosa	35.1	1.7
10	Chronic appendicitis	32.6	2.1

Figure S7 (a) Ten years old FFPE samples; (b) The Ct value of STEM-PCR using the mechanism described in Figure 1 targeting Septin9, provided with pathology description in (c).

We have added a statement on these results, accordingly:

“Further, to test the feasibility of STEM-PCR with lower quality DNA, we processed 10 formalin-fixed paraffin-embedded (FFPE) samples which were >10 years old. After extraction with a commercial kit (TAOGEN Inc., yielding concentrations 10.5 - 29.5 ng/μl; 260/280 ratio 1.94 - 2.12), results from STEM-PCR enabled differentiation between cancer types (see Figure S7), illustrating the potential for the technique to be used in retrospective studies, and with a wide range of samples.”

To further demonstrate clinical applicability (see also R1.0 from Reviewer 1), we extracted cfDNA from 14 patients and compared outcomes of testing using co-methylated STEM-PCR and standard bisulfite sequencing (Table S8), demonstrating similar performance between the two methods.

We added the following paragraph to discuss these results, and included the samples in the Methods Section, accordingly:

“To evaluate the clinical utility of the mechanism of co-methylation detection described in Figure 5, cfDNA extracted from 14 patients was tested using co-methylated STEM-PCR, and compared with gold-standard bisulfite sequencing, with the result shown in Table S8. Two samples (12, 13) showed positive with STEM-PCR (Ct value 27.86 and 25.25, equivalent to ca. 4 and 17 copies). The methylated results from bisulfite sequencing (Apogenomics Biotech CO Ltd, Shanghai, China) are also presented as methylation haplotype load²⁹, which was 1.89% (12) and 5.49% (13), whilst all

other samples had measures below 0.5% (considered as negative, following the kit recommendations). STEM-PCR matched the performance of standard bisulfite sequencing, but with only 1 ng ctDNA and no bisulfite pretreatment.”

Table S8. The results of STEM-PCR and Bisulfite sequencing using cfDNA. Samples 1-9 are healthy controls, samples 10-14 are patients diagnosed with colorectal cancer. In addition to patient-derived samples (1-14), we also included different proportions (10% and 1%) of fragmented methylated DNA with 30 ng unmethylated genomic DNA background to benchmark our bi-sulfite sequencing performance to the wider literature.

Sample	cfDNA concentration (ng)	STEM-PCR		Bisulfite sequencing	
		input cfDNA (ng)	Ct value	input cfDNA (ng)	Methylation haplotype load (%)
1	21.1	1	N	16.1	0.07
2	25.9	1	N	20.9	0.10
3	56.2	1	N	31	0.06
4	54.2	1	N	31	0.07
5	31.2	1	N	26.2	0.05
6	36.3	1	N	31	0.07
7	47.4	1	N	31	0.04
8	27.1	1	N	22.1	0.05
9	27.4	1	N	22.4	0.04
10	79.6	1	N	31	0.43
11	50.2	1	N	31	0.13
12	44.6	1	27.86	31	1.89
13	41.8	1	25.25	31	5.49
14	37.7	1	N	31	0.05
10%				30	21.28
1%				30	3.01
0%				30	0.15

N: no amplification

Q4.3. Following comment 2, STEM-PCR has only been tested on clinical tissues, while most epigenetic biomarker assays aim to detect methylation in liquid biopsies as this is perceived to be more clinically useful for most applications. STEM-PCR has been tested on simulated ctDNA (fragmented) but this is not the same as ctDNA (or cfDNA) from clinical liquid biopsies that could contain even smaller fragments and are of lesser quality. So, before statements on clinical applications can be made, the technique should be tested on ctDNA or cfDNA from clinical liquid biopsies, preferably liquid biopsies from archives or biobanks and not only on simulated ctDNA.

R4.3: We believe that we have now covered both of these points (see R4.2 and R4.3) as we believe that they are also related to clinical applicability. We use cfDNA samples for the demonstration of clinical use of STEM-PCR, and importantly, we also show similar analytical performance when compared to gold-standard bisulfite sequencing.

Q4.4. The authors indicate that STEM-PCR would be suitable for a broad range of applications in epigenetic research, which is indeed probably the case. It is also stated that this technique has potential for large-scale cancer screening, and in a clinical biomarker testing setting. This latter application is more challenging in my opinion, given the number of steps and modifications that need to be planned in the approach and the high number of custom/tailored components that are necessary to complete STEM-PCR. The technique appears to be quite complex due to these number of steps and tailored components, and therefore this might be too difficult to implement in clinical practice. At the moment, established high throughput detection systems are not always present in clinical institutions and some institutions lack the expertise to introduce these systems. STEM-PCR in itself can probably not be implemented at these sites, and even if integrated in established systems or in a commercially available kit, specific training will probably be necessary as there are many steps in the technique where problems or errors could occur. In addition, the technique is currently far from translation into a commercial kit, as it will need a lot of validation work on clinical samples of all sorts (see also comment 2). The message of the potential clinical application of this technique should therefore be toned down in the manuscript in my opinion.

R4.4: We agree with the Reviewer that applicability in broad clinical settings will require further developments (see also R4.1, above). We apologise for being too enthusiastic in our phrasing, although we only referred to compatibility and not directly to translation. We have limited the extent of our messaging accordingly. The important point that we wish to convey is the potential for other applications beyond methylation, as well as, in the future, the possible inclusion of the technique into an automated workflow.

“Described here for methylation detection, STEM-PCR also has potential for use in large-scale cancer screening, by, in future, integrating with high throughput automatic nucleic acid detection system, as a designed assay towards a set of targets, into an automated workflow (the technique is compatible with conventional PCR infrastructure).”

Q4.5. Following comment 2 and 3, it can be questioned whether STEM-PCR is not much more clinically valuable to detect mutations, instead of methylation. In the case of known mutations, will it not be much more easier to design a commercial kit specifically targeted at that known mutation? Easier than designing a commercial kit for methylation detection?

R4.5: We thank the Reviewer for highlighting the broad capability of STEM-PCR beyond methylation detection to also detect mutations, as shown in Figure 4 using PNA. It could indeed be argued that it would be simpler to design specific assays (through PNA sequence) to target specific mutations, once known. We note that the PNA modified blocker does not compete with TFP, which improves the design flexibility, whilst the artificial primer-dependent amplification of STEM-PCR requires only a small number of primers during the amplification step, which reduces design constraints for multiplex detection.

However, as highlighted in the manuscript (and discussed in previous comments), methylation detection using STEM-PCR is more practical than current gold standard BS-PCR with improved sensitivity and specificity (including low risks for false-positive), whilst MDRE-dependent digestion also retains sequence information.

We would argue that the discussion as to which application is best to strategically take forward in clinical validation will depend on an individual researcher's interest and we do not wish to restrict such endeavor in our text.

We do not believe that the points of the discussions around methylation detection need further emphasis in the text, but we have added a short note to highlight design flexibility for mutation detection as follows:

“Furthermore, since the PNA blocker does not compete with TFP, STEM-PCR offers extensive design flexibility to target different mutations”

Q4.6. No additional information is given on how to design or select the custom/tailored components that are necessary for STEM-PCR. Are tools to help the user available for this? Also for this part, specific training seems necessary before users can work with STEM-PCR.

R4.6: We agree with Reviewer’s opinion that an automated design tool would be valuable for the translation of the technique into the clinic or for wider application. In this study, we used the online IDT oligoAnalyzer tools ([Oligo Analyzer \(idtdna.com\)](https://www.idtdna.com/oligoanalyzer)) to avoid the formation of hairpin structures in TFP and dimers, in order to obtain a high amplification efficiency.

We have added a reference to this tool in the methods section:

“The online IDT oligoAnalyzer tools (<https://eu.idtdna.com/pages/tools/oligoanalyzer>) were used to avoid the formation of hairpin structures in TFP and dimers, in order to obtain a high amplification efficiency.”

Although we are now working on the development of an integrated software tool to provide candidate sequences (as outputs) for specific targets, as input, (see our previous work Xu et al., Nat Commun 13, 1635 (2022)), we believe that the details and examples provided in this current manuscript are sufficient to enable researchers with technical expertise in nucleic-acid based assays to design new sequences.

Q4.7. Methylated SEPT9 is used as an example to show the potential of STEM-PCR in biomarker research. For this marker, a commercial assay has long been available (EpiProColon). For a clinically useful comparison with STEM-PCR you would therefore expect a comparison with this commercial assay, why was this assay not used?

Q4.8. STEM-PCR is currently only compared to techniques that are similar to STEM-PCR. It would be interesting to see the comparison with other available (commercial) kits, to compare performance.

R4.7-8: We initially considered using EpiProColon, but we were not able to obtain details on sequences and locus, from the manufacturer. We instead opted to use the methylated SEPTIN9 detection kit from Biochainbj (Beijing, China), which has this information available. It is approved by NMPA and used in the clinic routinely. Figure S6b shows the sensitivity is up to 20 times better using STEM-PCR.

To evaluate the clinical utility of the mechanism of co-methylation detection described in Figure 5, we compared our STEM-PCR technique with standard bisulfite sequencing, using cfDNA extracted from 14 patients (Table S8) – see previous comment R4.2.

Q9. Figure 6 in the supplemental files shows the results of Sanger sequencing, depicting an almost perfect picture of sequencing results. Were these results actually derived from clinical samples as suggested in the text? In our hands, we have never seen such a perfect sequence chromatogram in clinical tissue samples as there always is a mixture of normal unmethylated and tumour methylated

tissue. Using this approach, you could miss the methylated allele as there is an abundance of unmethylated alleles. To overcome this, bisulfite sequencing of cloned alleles could be used as this would provide a more quantitative and more sensitive result. This approach could in fact lead to the picture as shown in Figure S6, but it has not been described in the materials and methods that this approach was used and that cloned alleles were sequenced. What is the approach used here, which results are shown in S6? If BS-seq is not performed on clones, the technique is much less specific, STEM-PCR should therefore be compared to BS-seq on cloned alleles to better illustrate the comparison of the two techniques.

R4.9: We agree with Reviewer’s point on the fact that in clinical samples there always is an “unmethylated” background. To alleviate potential concerns, we now provide some typical results from clinical samples processed either with BS-PCR or MSP-PCR sequencing (Figure S8). The quantitative results of DNA methylation can be analyzed as described by Jiang et al. (Lab Invest. 2010, 90: 282–290) (reference 21).

To further improve the sensitivity of BS-PCR, the samples with low levels of methylation were further amplified by methylated specific primer, after being diluted 10,000 X (Figure S8b). Three results shown mCpG information was missing with BS-PCR sequencing missed 3 mCpG, which were available present with MS-PCR sequencing. We used the protocol and steps of BS-PCR as suggested by reviewer, and compared outputs to STEM-PCR. We also added (reference 21) (Lab Invest. 2010, 90, 282–290.), which outlines the process of using the relative peak heights in direct bisulfite PCR sequencing traces to obtain quantitative results.

We added the following note in the results section to ensure that the process is clearer.

“To clarify the robustness of these results, we further increased the sensitivity of BS-PCR, by amplifying the samples with low levels of methylation (after being diluted 10,000 X, Figure S8). Typical results from clinical samples [Lab Invest. 2010, 90, 282–290] sequencing for SEPTIN9 and SFPR2 confirm the limitations in the performance of BS-PCR (Figure S8).”

Figure S8. Typical BS-PCR and MS-PCR sequencing results. (a). BS-PCR sequencing results of samples 5 and 19. (b). The BSP and MS-PCR results of sample 3.

Q4.10. Line 446: Inconsistent results were diluted 10,000 times and served as template for nest MSP to generate the methylated specific-PCR product for a second round of sequencing.

gttgtagggatgggttgaattgttgCGttttaggagaagtCGgggttggttaagaggaagagttCGCGCGtCGagtttaggt
agtagtgCGaggCGaggaagagtagtagtagCGagtttagggttttagtatCGtgggCGCGCGatttCGagggggtagag
ggagCGgagtCGgggaaggCGaggCGgtCGgagtCGagttgtttCGggttCGttttttCGttgggtgCGattCGgggtt
tCGaaaagttgtagtCGgCGgttggggCGCG

Primers are not properly marked in the table, and it is not specified which primers are used for which technique, but it can be assumed that the blue/bold primers are for BS and the yellow/italics primers for MSP. However, if you would dilute the blue product 10,000 times, the yellow product cannot be amplified. Is the description given in the manuscript correct? Have primers been depicted correctly? This should be checked.

R4.10: We thank the reviewer for this scrutiny and pointing out this mistake. We put the reverse primers of MSP-PCR and BS-PCR in the wrong position. The correct primer combination has been listed in Table S4. Thank you very much for this great attention to detail.

BS-PCR

SFRP2PF1 **GGAGAAGTAGGGTTGGTTAAAGAG**

SFRP2PR1 **CTACCAACTTTTCGAAACCC**

MS-PCR

SFRP2PF2 *GTTTAGGTAGTAGTGCGAGGC*

SFRP2PR2 *ACTCCGCTCCCTCTACCC*

Minor comments

Q4.11. Figure 1a, number 2, extension blocker is depicted below the figure, is this correct?

R4.11: To make the mechanism clearer, we moved the position of the extension blocker and added the 5' and 3'.

Q4.12. Numerous abbreviations are being used in the article and even in the figures, making it difficult to read. A list of abbreviations would be helpful.

R4.12: We prepared a list of abbreviations and added before the reference, and added this section after Data Availability. We will work with the editorial team to ensure the proper structure for the final manuscript is carried out. We would also consider including this in the ESI.

Specific Terminal Mediated Polymerase Chain Reaction	STEM-PCR
Allele-specific (AS) quantitative (q)PCR	AS-qPCR
Restriction enzyme	RE
PCR-restriction fragment length polymorphism	PCR-RFLP
Methylation sensitive restriction enzyme	MSRE
Helper-dependent chain reaction	HDCR
Methylation-dependent restriction endonuclease	MDRE
Peptide nucleic acid	PNA
Tailored-designed foldable primer	TFP
Capture region	CR

Artificial primer	AP
Folding region	FR
Colorectal cancer	CRC
Methylation-specific PCR	MS-PCR
Bisulfite-PCR	BS-PCR
Bridge primer	BP
Secreted Frizzled Related Protein 2	SFRP2

Q4.13. In line 217 the authors reference article [19] which describes reduced representation bisulfite sequencing libraries for genome-scale DNA methylation profiling; a technique for whole genome sequencing. In the Materials and Methods line 436, the authors refer to Sanger sequencing which is a different technique. Reference 19 does not refer to the standard Sanger sequencing as is implied in the text, is this reference correct?

R4.13: DNA methylation can be quantified using the relative peak heights in direct bisulfite PCR sequencing traces (Lab Invest. 2010, 90, 282–290.) - Reference 21.

Q4.14. No references are used when explaining the MSP methods, is this correct? Would be better to add some references.

R4.14: We added (Methods Mol Biol. 2011, 791:23-32) as reference 20 to support the MSP method.

Q4.15. Figure 5, the red dotted box is not visible in the figure.

R4.15: We increased the size of the line and used white to increase contrast.

Q4.16. Figure 1 from the supplemental file should be moved to the main text.

R4.16: We have integrated the Figure S1 with Figure 1.

Reviewer #5 (Remarks to the Author):

The manuscript by Xu et al entitled "Sequence terminus dependent PCR for site-specific mutation and modification" describes a new approach for detection of alterations in DNA sequence or DNA methylation. This method can assess single nucleotide changes (mutations) via a blocking oligonucleotide, or methylation sensitive restriction endonucleases to allow detection of methylation changes at a single CpG site.

The manuscript describes the approach well, and while a complicated approach to detect such changes, carefully demonstrates utility primarily on artificial templates mixed in unmodified DNA to demonstrate sensitivity of this approach. The primary advantage of this method for methylation detection is that unlike other restriction endonuclease methods which rely primarily on the absence of cleavage to detect methylation, this method only detects methylation when methylated sequences are cleaved (and thus the incomplete cleavage of previous methods generating a false positive is eliminated). It is possible that star activity of the RE could produce a false positive and might be discussed, but may be minimal for these endonucleases.

For mutational detection, a blocking oligo prevents the necessary structure for amplification.

Detailed figures in the manuscript show the potential use of such an approach, which is described as providing single nucleotide resolution of DNA methylation or mutation without sequencing, and doing so with greater sensitivity than sequencing.

While these advantages are nicely described and shown, the limitations of STEM-PCR are not mentioned.

Q5.1 First, the limitation for methylation detection to examination of single CpG sites requires these lie in a methylation sensitive RE recognition sequence. Other sites would not be amenable to examination. In addition, the requirement to design a specific assay for each CpG site is also complex, rather than the ability to determine DNA methylation status at multiple sites within a sequenced region. In this latter consideration, the detection of co-methylation is demonstrated in figures 5 and 6, we an assay using two methylation sensitive RE recognized sites to detect concordant methylation at these two sites, but this would not discriminate unmethylated from sequences methylated at either of the two sites, which would I believe require use separate digestion of template with only one RE to detect partial methylation, greatly complicated the analysis. In contrast, bisulfite sequencing or alternative modification sequencing approaches recently developed provide individual allele patterns of the 4 possible states, as well as the status of adjacent CpG sites.

R5.1: We thank the Reviewer for their positive comments and constructive suggestions. Comments from all reviewers have enabled us to provide specific discussions on the benefits and limitations of the technique in different parts of the manuscript (e.g. R1.5 led to a discussion on the opportunity of using PNA sequences and hybridization limitations). More specifically here, as discussed in R4.1, we also agree that the technique currently requires the biomarkers to be established so that the appropriate assay is designed. In the case of co-methylation, the example provided in the manuscript highlights the ability of STEM-PCR to detect such challenging scenarios, where co-methylation detection increases the specificity significantly compared to a single mutation. However, as noted by the reviewer, the assay design does not differentiate between the methylation sites independently.

We do believe that the manuscript now contains more balanced statements so we did not add a dedicated part summarizing these extensively. However, we think that a brief section to conclude could be useful for the reader and we added the following sentence at the end of the manuscript integrated with the points required by R4.1:

"It should also be noted that the capability of STEM-PCR for location-specific methylation detection depends on the specificity of the digestion by MDREs, which can be limited for some methylated CpG and does not allow the differentiation of single mutations in co-methylation analysis"

Q5.2 The sensitivity of this approach is shown, and in supplemental data reported to be 20 times more sensitive than Heavy Methyl PCR, which is stated in the manuscript to thus be 20 fold more sensitive than the state of the art. Heavy Methyl is a difficult approach only developed for commercial use for IP issues, and is not the most sensitive approach for methylation detection. As shown in the data in figure 3, while STEM-PCR was more sensitive than sequencing (expected), the confirmation Methylation Specific PCR then assessed for this site methylation by sequence verification was equal to STEM-PCR. The reported sensitivity of 0.1% methylation in the manuscript was reported in the original MSP manuscript, with realtime MSP showing greater sensitivity. Other newer approaches have also shown much greater sensitivity than 0.1% and reach single molecule detection. Figure 3 does show reasonably sensitive detection, but doesn't provide any confirmation that this is specific

(ie no control patients were included). This is important to assess, since the detection of single site methylation (which is the focus of STEM_PCR) may not provide the level of specificity needed for this clinical application.

R5.2: This comment is consistent discussions from Reviewer 4 (see 4.9). Briefly, we used the total methylated and unmethylated genomic DNA as a control for the standard BSP and MSP sequencing. We show typical clinical results of BS-PCR sequencing in Figure S8a. To further improve the sensitivity of BS-PCR, the samples which were deemed unmethylated were further amplified by methylated primer after 10,000X dilution (Figure S8b). Importantly, we now show results from a direct comparison using cfDNA extracted from patients (Table S8), which includes “control” patients.

Although a small sample cohort, we believe that the new results from clinical samples provide the confidence of the technique’s performance compared to benchmark.

Q5.3 The mutational detection of STEM-PCR is demonstrated for EGFR detection of the common L858R mutation in figure 4. Again, showing detection down to 30 copies, this is a sensitive detection approach and is shown to be specific. The limitations of this analysis to just a single mutation in EGFR is not discussed, and such an approach would require a specific assay for each of the known point mutations in this gene, and would not readily I believe detect the deletion mutations also common (this may not be true, if the PNA would cross a deletion site, but was not discussed and would also require different PNAs for each potential deletion).

R5.3: The reviewer is correct and we apologise for any misunderstanding. In STEM-PCR, the PNA modified blockers need to hybridize to the deletion sequences. Thus, for each deletion, specific PNA-modified blockers or specific TFP, with specific FRs should be designed.

We have added a statement at the end of Figure 4.

“However it should be noted that each mutation requires its own set of PNA blockers.”

And emphasized it in the Discussion Section.

“Furthermore, for each deletion, a specific set of PNA-modified blockers or specific TFP and FRs need to be designed.”

REVIEWERS' COMMENTS

Reviewer #1 (Remarks to the Author):

The authors have addressed all my previous concerns and comments. I thank them for performing the additional experiments that give a better insight in the performance and robustness of the method. I have no other remarks.

Reviewer #3 (Remarks to the Author):

The manuscript entitled "Sequence terminus dependent PCR for site-specific mutation and modification detection" describes a new approach for detection in DNA methylation. The authors gave a detailed reply to the comments of the reviewers. However, the paper seems more suitable for technical journals.

Reviewer #4 (Remarks to the Author):

The authors addressed all comments, and the paper was significantly improved.

Reviewer #5 (Remarks to the Author):

The authors have responded to the issues raised by the panel of reviewers. Additional experiments have been performed to address concerns of the lack of biologically relevant specimens rather than artificial templates to demonstrate this approach. The STEM pcr approach described overcomes some limitations of many other approaches. Specifically, it can assess rare changes, either mutation or methylation following modification, at defined sites. The authors compare this to sequencing where this eliminates some issues of sequencing errors and by utilizing pcr brings sensitivity not possible with typical sequencing depths. While the authors do provide modified text to define the very specific nature of the assay design, the limitations to a relatively small number of point mutations and site specific methylation remains and may limit utility. The examples of colon tissue S7 and in table s8 for blood samples show some utility, but the blood detection in table s8 of only 2/5 cancers detected and only those with bisulfite sequencing levels of >1.5% does not demonstrate sensitivities of other approaches for detection.

Reviewer #1 (Remarks to the Author):

Q1a. The authors have addressed all my previous concerns and comments. I thank them for performing the additional experiments that give a better insight in the performance and robustness of the method. I have no other remarks.

And Reviewer #4 (Remarks to the Author):

Q1b. The authors addressed all comments, and the paper was significantly improved.

A1. We in turn thank the reviewers for their thorough suggestions that greatly improved the manuscript.

Reviewer #3 (Remarks to the Author):

Q2. The manuscript entitled "Sequence terminus dependent PCR for site-specific mutation and modification detection" describes a new approach for detection in DNA methylation. The authors gave a detailed reply to the comments of the reviewers. However, the paper seems more suitable for technical journals.

A2. We thank the reviewer for acknowledging the efforts made in revising the manuscript. We respectfully disagree with the reviewer on the unsuitability of the work for Nature Communications. The work demonstrate a new technique to study nucleic acid sequences which has the potential to impact a broad range of applications, including in the clinic. We strongly believe that this will benefit the wide readership of the journal.

Reviewer #5 (Remarks to the Author):

Q3. The authors have responded to the issues raised by the panel of reviewers. Additional experiments have been performed to address concerns of the lack of biologically relevant specimens rather than artificial templates to demonstrate this approach. The STEM pcr approach described overcomes some limitations of many other approaches. Specifically, it can assess rare changes, either mutation or methylation following modification, at defined sites. The authors compare this to sequencing where this eliminates some issues of sequencing errors and by utilizing pcr brings sensitivity not possible with typical sequencing depths. While the authors do provide modified text to define the very specific nature of the assay design, the limitations to a relatively small number of point mutations and site specific methylation remains and may limit utility. The examples of colon tissue S7 and in table s8 for blood samples show some utility, but the blood detection in table s8 of only 2/5 cancers detected and only those with bisulfite sequencing levels of >1.5% does not demonstrate sensitivities of other approaches for detection.

A3. We thank the reviewer again for their detailed study of the new results provided. We are grateful for the emphasis on the fact that our PCR approach brings significant advantages when compared with DNA sequencing methods. We also acknowledge the fact that contrary to sequencing, our method can only target a limited number of points in sequences of interest. However, as demonstrated in the results on patient samples (pointed out by the reviewer), the method has the potential to be used easily in clinical contexts.

We disagree with the reviewer's comment that the results on Table S8 show limited sensitivity. Indeed we demonstrate that our method is able to detect at least the same range of patients than the gold standard BS-sequencing, but using significantly less source material, providing the potential for the technique to perform better when material is limited in future clinical studies. Unfortunately, the samples available did not allow us to establish a limit of detection in clinical applications, however our analytical characterisation (Figure 3) demonstrates that STEM-PCR is potentially more sensitive than BS-seq. We have added a further statement to the discussions of Table S8 in the main text, as follows:

“STEM-PCR matched the performance of standard bisulfite sequencing, but with only 1 ng ctDNA and no bisulfite pretreatment, confirming the potential of the technique to perform advantageously in applications where source material is limited. However more extensive clinical evaluation will be required in future, to establish clinical limit of detection”